psychology

stimulation, self-sufficiency, materialism, social, input

**Author for correspondence:**
Dario Krpan
e-mail: d.krpan@lse.ac.uk

# Exploring the need for external input through the prism of social, material and sensation seeking input

## Dario Krpan

Department of Psychological and Behavioural Science, London School of Economics and Political Science, Houghton Street, London WC2A 2AE, UK

DK, 0000-0002-3420-4672

External input is any kind of physical stimulation created by an individual's surroundings that can be detected by the senses. The present research established a novel conceptualization of this construct by investigating it in relation to the needs for material, social and sensation seeking input, and by testing whether these needs predict psychological functioning during long- and short-term input deprivation. It was established that the three needs constitute different dimensions of an overarching construct (i.e. need for external input). The research also suggested that the needs for social and sensation seeking input are negatively linked to people's experiences of long-term input deprivation (i.e. COVID-19 restrictions), and that the need for material input may negatively predict the experiences of short-term input deprivation (i.e. sitting in a chair without doing anything else but thinking). Overall, this research indicates that the needs for social, material and sensation seeking input may have fundamental implications for experiences and actions in a range of different contexts.

## 1. Exploring the need for external input through the prism of social, material and sensation seeking input

'All of humanity's problems stem from people's inability to sit quietly in a room alone.' Blaise Pascal, *Pensées*.

One of the primary characteristics of human beings is that they have difficulties maintaining optimal functioning in impoverished environments. Whereas spending short periods in spaces with reduced sensory input and engaging in meditation can be beneficial (e.g. [1]), prolonged periods without stimulation can lead to negative consequences for mental functioning and

wellbeing (e.g. [2]). In line with these considerations, it is plausible that human life itself is a continuous process of avoiding 'empty' environments and securing the input that is necessary for functioning. In this context, I define 'input' as any kind of physical stimulation created by the external environment that can be detected by the senses.

Input has many layers. It can involve basic sensory stimulation, such as touch, all the way to more complex forms such as someone's voice conveying a thought. In personality psychology, three separate research areas have focused on different aspects of input by examining people's needs for *social*, *material* and *sensation seeking* input. In this context, the need for any of these three forms of input reflects how much people require it to maintain optimal psychological functioning, broadly defined as the presence of positive and absence of negative affective states [3]. Social input refers to stimulation generated by contact with other human beings (e.g. [4]); material input to stimulation arising from money and material objects more generally (e.g. [5]); and sensation seeking input to any stimulation that is experienced as novel and/or intense (e.g. [6]).

The main objective of the present research is to achieve a fundamental psychological understanding of people's need for external input by approaching it from the perspective of these three research areas. I explore two key questions in this regard. First, are the needs for material, social and sensation seeking input related, and what is their link to the overarching construct of the need for external input (Research Question 1; RQ1)? Second, do the three needs shape psychological functioning during long- and short-term input deprivation (Research Question 2; RQ2)? In the next section, I first overview previous research on the link between the needs for input and then examine their consequences for psychological functioning during periods of input deprivation. Finally, I provide a brief overview of the present research before proceeding with each study.

## 2. The needs for social, material and sensation seeking input

Various individual difference constructs that, at least to some degree, tap into the three needs for input have been developed. Concerning social input, these constructs involve the need to belong [7]; fundamental social motives and affiliation motivation [4,8]; preference for solitude [9]; attachment [10]; dependency [11]; collectivism-individualism [12]; communal orientation [13]; loneliness [14]; aloneness [15] and relatedness [16]. Constructs that capture the need for material input comprise materialism and material values [5]; greed (e.g. [17]) and impulsive buying [18]. Finally, constructs that assess the need for sensation seeking input include sensation seeking (e.g. [19,20]); novelty/change seeking [21,22]; sensory processing sensitivity [23] and need for stimulation [24].

Considering that I approach the construct of external input through the prism of needs, it is necessary to situate it in relation to previous literature on needs. I do not conceptualize the needs for social, material and sensation seeking input as biological or physical needs that typically refer to the basic requirements for maintaining human life, such as food or shelter (e.g. [25,26]). Indeed, even if the expressions that I use in relation to external input, such as 'any physical stimulation that can be detected by the senses', may be evocative of physical needs, I use these expressions because I approach external input from a materialist perspective rather than because I write about physical needs. Given that I define the needs for social, material and sensation seeking input in terms of how much people require the corresponding input components to maintain optimal psychological functioning, these needs can be classified as experiential needs (i.e. needs that are important for wellbeing; [27]). However, based on previous research, it would be difficult to argue whether these three needs are basic psychological needs in the sense that they all need to be met for a person to experience high wellbeing, as would be the case for the needs such autonomy, competence and relatedness stemming from the self-determination theory [28]. Whereas investigating this would be an interesting topic for future research, it is not the focus of the present article.

The literatures studying the needs for social, material and sensation seeking input have led separate lives, presumably because researchers have not recognized the common denominator of 'input' that connects them. For this reason, it is difficult to infer whether the needs for social, material and sensation seeking input are associated and understand their link to the overarching construct of the need for external input. However, the limited evidence to some degree suggests that the three needs are positively related. For example, regarding the relationship between social and material input, the need to belong was positively related with materialism [29]. Regarding the relationship between material and sensation seeking input, materialism was positively related to sensation seeking [30].

Finally, concerning social and sensation seeking input, relatedness was positively correlated with the need for novelty [31].

## 3. Consequences of the needs for input during input deprivation

Whereas relatively few studies investigated the links between the needs for social, material and sensation seeking input, research examining how these needs shape psychological functioning during long- and short-term input deprivation is even less frequent. This is because, for ethical reasons, studying long-term input deprivation is possible only in the rare situations where it naturally occurs (e.g. self-isolating during a pandemic, or working in isolated and confined environments (ICEs) such as the polar regions or space; [32,33]), whereas situations of short-term input deprivation have generally been understudied, especially in relation to personality.

The limited evidence indicates that social and sensation seeking input has negative consequences for people's experiences of long-term input deprivation, whereas material input has not been examined in this respect. For example, in a study about people working in Antarctic stations, higher affiliation motives were associated with higher levels of loneliness, which in turn predicted higher frequency of negative versus positive moods and increased cognitive impairment [34]. Moreover, sensation seeking was positively correlated with the boredom experienced during COVID-19-related lockdowns [35].

Concerning short-term input deprivation, the limited evidence suggests that material and sensation seeking input have negative consequences for people's experiences. For example, early research showed that people higher (versus lower) in need for stimulation were more likely to actively stimulate themselves by observing visual images during short-term sensory isolation [36], and they also perceived such periods as more disturbing [37]. Moreover, people's daily smartphone use predicted lower enjoyment during a 10–15 min period of restricted stimulation [38]. Although smartphone use is not a measure of material input, it is typically associated with materialism (e.g. [39]), and this finding may therefore indicate that higher need for material input predicts lower wellbeing during short-term input deprivation.

## 4. Main conclusion and overview of the present research

Overall, although the needs for material, social and sensation seeking input are conceptually related because they reflect how people experience different aspects of sensory stimulation, few studies probed the link between the three needs, and no research investigated these needs in relation to the overarching construct of the need for external input (RQ1). Similarly, no study systematically investigated the relative importance of the three needs for psychological functioning during long- and short-term input deprivation (RQ2). The research questions at the core of the present article therefore remain unanswered.

To address RQ1, in Study 1, I first developed a scale to measure individual differences in the needs for external input. Developing a new scale was necessary because the existing scales were created separately for each of the three needs and contain different language styles, combine wordings that mix preferences and needs and, in addition to measuring input, typically include questions that tap into themes beyond input (for an extended discussion of these issues, see electronic supplementary material, pp. 6–8). Using these scales, avoiding various confounds when comparing the needs for social, material and sensation seeking input would thus be difficult. In Study 2, I extended the answer to RQ1 by probing convergent and discriminant validity of the new scale. Finally, in Studies 3 and 4, I examined RQ2 by testing whether material, social and sensation seeking input predict emotional experiences and behaviour during long- and short-term input deprivation.

## 5. Study 1: the need for external input scale

Study 1 tackled RQ1 by investigating the needs for social, material and sensation seeking input in relation to an overarching construct—the need for external input. For this purpose, I first devised a set of items assessing the three input components and then subjected them to exploratory factor analysis (EFA),

which provided preliminary answers to the research question. Subsequently, I conducted confirmatory factor analyses (CFAs) to directly answer the question. More specifically, the analyses fitting second-order confirmatory factor models probed whether the material, social and sensation seeking factors load highly (i.e. greater than or equal to 0.50; [40]) on the overarching factor of external input and therefore reflect different dimensions of this construct [41,42]. Moreover, fitting bifactor models allowed further investigating whether the construct is in fact multi-dimensional (i.e. it comprises these three factors), or it is uni-dimensional, and it can therefore be reduced to only one factor [43,44]. This would indicate that, even if on a conceptual level material, social and sensation seeking input appear to measure different aspects of the need for external input, they in fact capture the same overlapping information about the construct.

## 5.1. Methods section

For this and other studies in the article, all manipulations, measures and exclusions are reported. No studies were preregistered. Complete materials, data, analysis codes and codebook for interpreting the data files for all studies are available via the Open Science Framework (OSF): https://osf.io/mxezt/?view_only=4f65da59314443a89bd91156b2ae5ec2.

### 5.1.1. Item development

Thirty items were developed: 10 for each input component (table 1; items 1–10: material input; items 11–20: sensation seeking input; items 21–30: social input). The items had to satisfy several criteria. First, given that, as discussed more comprehensively in the electronic supplementary material (pp. 6–8), one of the main shortcomings of the existing measures is that they contain items that tap into themes beyond the need for input (e.g. whether people like to be surrounded only by others impressed by them; [45]), all items had to capture specifically input (e.g. items 25 & 17, table 1). Second, because the existing scales combine items that capture both preferences and needs (e.g. 'I like a lot of luxury in my life' versus 'I have all the things I really need to enjoy life', [5, p. 217]; see also electronic supplementary material, pp. 6–8), all items were phrased more strongly to go beyond mere preferences and tap into a need. This means they had to either refer to a need (e.g. item 3, table 1) or be phrased in terms of some form of dependency between a specific input component and the absence of negative or presence of positive affective states (e.g. item 16, table 1).

Third, to identify the terms that denote different input components in each of the items developed, I reviewed the labels that other commonly used scales used for this purpose (electronic supplementary material, pp. 9–13). Therefore, terms material things, material objects and money were used in reference to material input; terms stimulating activities, stimulation, new sensations, stimulation, new experiences, novel activities and doing many different things at once in reference to sensation seeking input; and terms other people, friends and people who are close to me in reference to social input. Finally, because I defined the need for external input in terms of how much people need a specific type of input (i.e. material, sensation seeking or social) to maintain optimal functioning, operationalized as the presence of positive or absence of negative affective states [3], the words denoting affective states were selected based on a review of such words typically employed by other scales that tackle input (electronic supplementary material, pp. 14–17). Therefore, the following terms were used to label positive affective states: happy, excited/excitement, energized, feel good, good time, enjoy, like, fun, satisfied, amuse, pleasure, elated and joy. For negative affective states, I used the terms unhappy, boredom/bored, upset, irritated, unbearable, half-dead, anxious, dull, uncomfortable, miserable and meaningless.

The items were scored using a 7-point Likert scale from '1 = Strongly disagree' to '7 = Strongly agree' due to its desirable psychometric properties [46]. I did not use reverse-worded items; all items were phrased in the same direction, with higher scores indicating higher need for input. Although reverse-worded items are sometimes used to reduce acquiescence bias, they can create various methodological issues (e.g. [47]). Moreover, they do not effectively tackle this bias, and a more successful approach involves designing short scales with roughly 10 items phrased in the same direction [48]. Therefore, for the final version of need for external input scale (NEIS) that was subjected to CFA, I aimed to select 12 items, four for each factor.

**Table 1.** Factor loadings (standardized), variance explained, eigenvalues and correlations among factors from the EFA performed on Sample 1 (Study 1). *Note*. Labels F1–F4 refer to Factors 1–4, respectively. Items in italics were used in the final version of the need for external input scale (NEIS) validated in the CFAs, and their numbers in parentheses correspond to the numbers used in the final scale version reported in figure 1. Only factor loadings greater than or equal to 0.32 are reported for clarity. Coefficients for Factors 1–4 at the bottom of table 1 denote correlations between the factors. Items that have highest loadings on Factor 1 all correspond to social input, those that have highest loadings on Factor 2 correspond to material input, and the ones with highest loadings on Factor 3 to sensation seeking input.

| item no. | F1 | F2 | F3 | F4 |
|---|---|---|---|---|
| 28. I feel miserable if I don't frequently interact with other people. | 0.871 | | | |
| 29. Without other people, my life would feel meaningless. | 0.841 | | | |
| *26. (5.) I need to frequently see or talk to people who are close to me to be happy.* | 0.805 | | | |
| *25. (1.) To have fun, I need to be with other people.* | 0.734 | | | |
| *30. (9.) I can experience true joy only when I am surrounded by people close to me.* | 0.727 | | | |
| *27. (10.) If I spend a lot of time without other people, I feel uncomfortable.* | 0.623 | | | |
| 24. The extent to which I can enjoy my life depends on how many friends I have. | 0.618 | | | 0.326 |
| 23. I need other people to escape boredom. | 0.607 | | | |
| 21. To feel excitement, I need to be surrounded by other people. | 0.607 | | | |
| 22. I can have a good time only if I am in the company of other people but not if I am alone | 0.585 | | | 0.373 |
| *10. (3.) I feel irritated if I can't buy material objects I desire.* | | 0.855 | | |
| 6. I get upset if I can't afford any material things that I like. | | 0.825 | | |
| 9. I need material objects to amuse myself. | | 0.779 | | |
| 8. If I can buy material things, I am satisfied. | | 0.761 | | |
| *3. (7.) I need material objects to avoid boredom.* | | 0.703 | | |
| 5. If I have very little money to spend I can't enjoy my life. | | 0.681 | | |
| *4. (11.) I need attractive material things to feel good.* | | 0.665 | | 0.326 |
| *7. (12.) The extent to which I can have fun depends on how much money I can spend.* | | 0.664 | | |
| 1. If I can't spend time in places with a lot of material things, I start feeling unhappy. | | 0.511 | | 0.369 |
| 2. I cannot feel excited unless I am surrounded with many material objects. | | 0.504 | | 0.462 |
| *16. (6.) Without new experiences, I feel half-dead.* | | | 0.931 | |
| *14. (8.) When I have not experienced new sensations for longer periods of time I start feeling unhappy.* | | | 0.800 | |
| 13. I constantly need new sensations to avoid being bored. | | | 0.733 | |
| *19. (2.) If my senses are not stimulated, I feel dull.* | | | 0.705 | |
| *17. (4.) To me, pleasure is about new sensations and experiences.* | | | 0.677 | |

| item no. | F1 | F2 | F3 | F4 |
|---|---|---|---|---|
| 11. If I can't frequently engage in highly stimulating activities my life seems unbearable. | | | 0.650 | |
| 18. I get anxious if I cannot frequently engage in novel activities. | | | 0.619 | |
| 12. To enjoy my life, I need more stimulation than other people. | | | 0.573 | |
| 15. I need a lot of stimulation to get excited and energized. | | | 0.535 | |
| 20. I feel elated only when I do many different things at once. | | | 0.373 | |
| variance explained | 19.238% | 19.440% | 18.824% | 6.454% |
| eigenvalues | 5.771 | 5.832 | 5.647 | 1.936 |
| F1 | — | | | |
| F2 | 0.551 | — | | |
| F3 | 0.741 | 0.654 | — | |
| F4 | 0.389 | 0.378 | 0.397 | — |

**Table 2.** Sample size and demographics for participants who were included in statistical analyses in Studies 1–4. *Note.* Studies were administered via Qualtrics. In Studies 1 (Samples 1 and 2), 2 and 4, participants were recruited via Amazon Mechanical Turk (MTurk). In Study 1 (Sample 3), participants were recruited via Prolific Academic. In Study 3, participants were recruited via social media, predominantly Facebook. For demographics and sample sizes of all participants recruited (i.e. including those who were not included in analyses), see electronic supplementary material (p. 4).

| study | sample no | sample size | *M* age | s.d. age | gender | | | country |
|---|---|---|---|---|---|---|---|---|
| | | | | | male | female | other | |
| 1 | 1 | 397 | 33.882 | 10.386 | 237 | 160 | 0 | US |
| 1 | 2 | 802 | 35.989 | 11.416 | 398 | 403 | 1 | US |
| 1 | 3 | 418 | 35.639 | 11.953 | 136 | 277 | 5 | UK |
| 2 | — | 317 | 37.233 | 11.469 | 166 | 151 | 0 | US |
| 3 | — | 1992 | 43.922 | 13.734 | 201 | 1767 | 24 | various[a] |
| 4 | — | 519 | 36.155 | 10.150 | 342 | 176 | 1 | US |

[a]People from the following countries took part in Study 3: Australia, Austria, Belgium, Brazil, Bulgaria, Canada, China, Colombia, Croatia, Denmark, Finland, France, Germany, Greece, Guatemala, Hong Kong, India, Ireland, Isle of Man, Italy, Japan, Kuwait, Lebanon, Lithuania, Luxembourg, Malaysia, Malta, Mexico, Netherlands, New Zealand, Norway, Philippines, Poland, Portugal, Romania, Serbia, Singapore, South Africa, South Korea, Spain, Sweden, Switzerland, Turkey, United Arab Emirates, United Kingdom and USA.

### 5.1.2. Participants and procedure

Three samples were collected: table 2 presents only participants included in statistical analyses, whereas the information about all participants and the exclusion criteria are detailed in the electronic supplementary material (pp. 4–5). The sample size rationale which was informed by various recommendations (e.g. [49]) and a power analysis based on Monte Carlo simulations [50] is also detailed in the electronic supplementary material (pp. 18–19). In each sample, participants first

received the consent form and then responded to the need for input items. In Sample 1, they received all 30 items that were developed, whereas in Samples 2–3, they responded only to the 12 items selected for NEIS (table 1). In the end, participants' demographics (gender, age and nationality) were assessed, and they were debriefed.

## 5.2. Results

### 5.2.1. Exploratory factor analysis

The analysis was conducted on Sample 1. I first computed the Kaiser–Meyer–Olkin measure of sampling adequacy, which was 0.968, and Bartlett's test of sphericity, which was significant, $\chi^2_{435} = 9518.021$, $p < 0.001$, thus indicating the data were suitable for EFA [51]. Moreover, a parallel analysis [52–54] suggested retaining four factors. I therefore computed a maximum-likelihood (ML) EFA with four factors (table 1) and used an oblique rotation—*geomin*—given that I expected the factors to be correlated because they capture an overarching theoretical construct and because this rotation typically produces a clean factor structure [55].

As indicated in table 1, Factor 4 explained a substantially smaller proportion of variance than the other factors and did not produce high loadings (i.e. greater than or equal to 0.50) that would warrant inclusion in a scale [56]. This factor was therefore discarded because it may be a result of overfactoring, to which parallel analysis is prone [57]. The retained Factors 1–3 corresponded to social, material and sensation seeking input, respectively, given that these factors had highest loadings for the items created to capture these input components. Correlations between the factors were also high (table 1), which supports preliminary indications from the literature review. For each of the retained factors, representative items that were further tested in CFA were selected based on several statistical and conceptual criteria (electronic supplementary material, p. 20).

### 5.2.2. Confirmatory factor analysis

A second-order CFA was performed, in which items were set to load on the first-level factors representing material, social and sensation seeking input, and these factors were set to load on the second-level factor representing the need for external input. This analysis yields identical fit indices as the first-order CFA, and yet it allows testing whether the three first-level factors load highly (i.e. greater than or equal to 0.50) on the second-level factor and therefore comprise its underlying dimensions [40,42]. To evaluate model fit, I used the following fit indices and cut-off values: standardized root mean square residual less than 0.08; comparative fit index (CFI) > 0.90 and root mean square error of approximation (RMSEA) < 0.10 [58,59]. Model fit was estimated using the maximum likelihood mean-variance adjusted (MLMV) ML estimator with robust standard errors [60].

As figure 1 shows, CFAs on Samples 1–3 revealed that the model had acceptable fit, and standardized loadings on the higher order factor were all high (greater than 0.595). Therefore, it was confirmed that the needs for social, material and sensation seeking input correspond to different dimensions of the need for external input. Next to confirming the factor structure, I showed that this model had a better fit than other possible models (electronic supplementary material, p. 20). Descriptive statistics for each first-level factor are available in table 3.

### 5.2.3. Probing uni-dimensionality of need for external input scale

To further probe whether the need for external input can be reduced to only one dimension despite the CFAs establishing three dimensions (material, social and sensation seeking input), I applied bifactor statistical indices [43]. According to Rodriguez *et al*. [44], a scale is uni-dimensional if the following criteria are met: omega hierarchical ($\omega_h$) greater than 0.80; factor determinacy (*FD*) greater than 0.90; construct replicability index (*H*) greater than 0.80; explained common variance (*ECV*) greater than 0.70 and percentage of uncontaminated correlations (*PUC*) greater than 0.70. For Sample 1, $\omega_h$, *FD*, *H*, *ECV* and *PUC* were 0.81, 0.91, 0.90, 0.68 and 0.73, respectively. For Sample 2, these indices were 0.71, 0.86, 0.84, 0.54 and 0.73. For Sample 3, they were 0.67, 0.85, 0.82, 0.51 and 0.73. Therefore, the cut-off criteria were not met for Samples 1–3, and there was no evidence in support of uni-dimensionality of the construct.

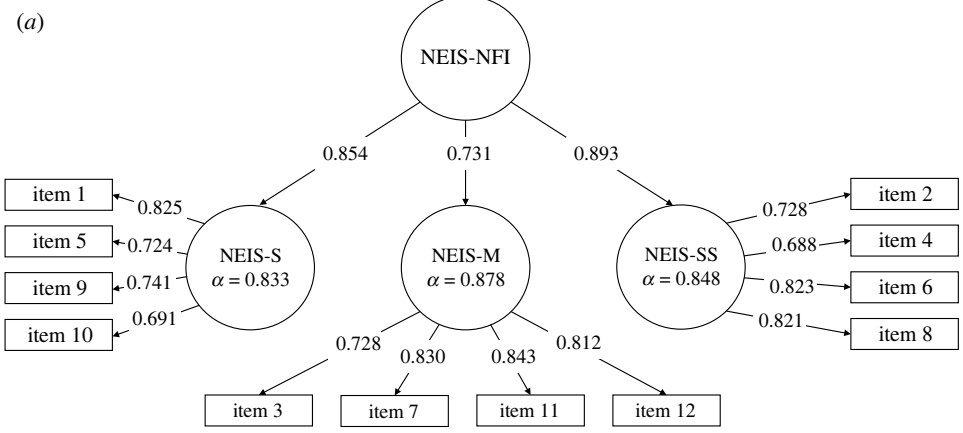

$\mathcal{X}^2(51) = 60.347, p < 0.174$; SRMR = 0.031; CFI = 0.994; RMSEA = 0.26, 90% CI [0.001, 0.050]

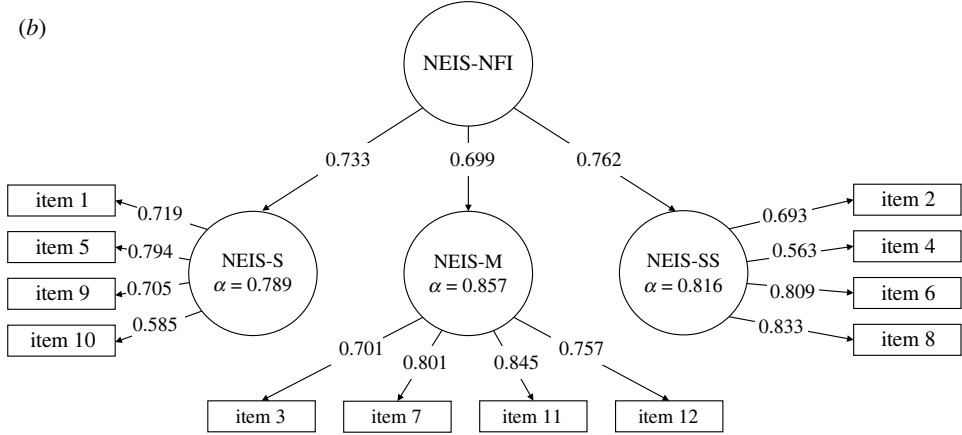

$\mathcal{X}^2(51) = 166.470, p < 0.001$; SRMR = 0.048; CFI = 0.957; RMSEA = 0.066, 90% CI [0.055, 0.077]

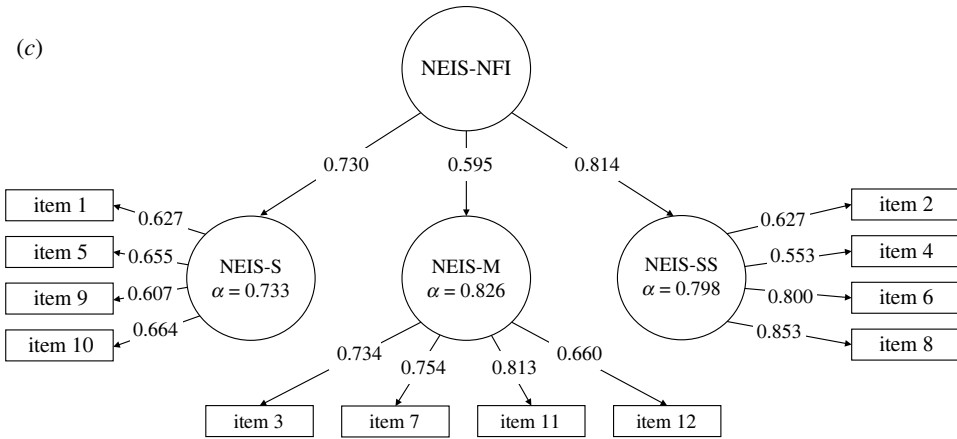

$\mathcal{X}^2(51) = 108.531, p < 0.001$; SRMR = 0.045; CFI = 0.954; RMSEA = 0.061, 90% CI [0.045, 0.077]

**Figure 1.** Factor structure of the NEIS evaluated using second-order CFAs on Samples 1 (*a*), 2 (*b*) and 3 (*c*). Indices RMSEA and CFI were computed based on recommendations by Savalei [61]. All item and factor loadings are standardized; for the content of each item, table 1. Coefficients $\alpha$ refer to Cronbach's $\alpha$s. NEIS-S indicates the need for social input; NEIS-M the need for material input; and NEIS-SS the need for sensation seeking input. NEIS-NFI corresponds to the need for input as the higher order factor. In Sample 1, the correlations between NEIS-S and NEIS-M, NEIS-S and NEIS-SS, and NEIS-M and NEIS-SS were $r = 0.529$, $r = 0.644$ and $r = 0.576$, respectively; in Sample 2, they were $r = 0.448$, $r = 0.469$ and $r = 0.451$; and in Sample 3, they were $r = 0.338$, $r = 0.477$ and $r = 0.398$ (all $ps < 0.001$).

**Table 3.** Descriptive statistics for the NEIS factors in Samples 1–3 in Study 1. *Note.* NEIS = need for external input scale; NEIS-S = NEIS social input; NEIS-M = NEIS material input; NEIS-SS = NEIS sensation seeking input.

| sample | *N* | min | max | *M* | s.d. |
|---|---|---|---|---|---|
| 1 | | | | | |
| NEIS-S | 397 | 1 | 7 | 3.678 | 1.472 |
| NEIS-M | 397 | 1 | 6.75 | 3.465 | 1.526 |
| NEIS-SS | 397 | 1 | 7 | 3.931 | 1.418 |
| 2 | | | | | |
| NEIS-S | 802 | 1 | 7 | 3.513 | 1.345 |
| NEIS-M | 802 | 1 | 7 | 3.255 | 1.406 |
| NEIS-SS | 802 | 1 | 7 | 3.995 | 1.291 |
| 3 | | | | | |
| NEIS-S | 418 | 1 | 6.75 | 3.706 | 1.227 |
| NEIS-M | 418 | 1 | 7 | 3.210 | 1.328 |
| NEIS-SS | 418 | 1 | 7 | 4.060 | 1.241 |

## 5.3. Discussion

Study 1 answered RQ1 by showing that the items capturing the needs for social, material and sensation seeking input load onto separate factors that are positively correlated, and that these factors constitute different dimensions of the need for external input as an overarching construct. The present study also led to the development of NEIS (table 1) that has good psychometric properties and can be used to measure individual differences regarding the three needs for input.

# 6. Study 2: convergent and discriminant validity

Study 2 had three objectives. First, I wanted to establish convergent validity of NEIS by showing the needs for social, material and sensation seeking input correlate with other measures of these needs. Considering that Study 1 showed the three needs constitute different dimensions of the need for external input, to further address RQ1 I also probed whether these correlations go beyond the corresponding needs (e.g. social–social) and occur across needs (e.g. social–sensation seeking). Second, I aimed to establish discriminant validity of NEIS by showing that it differs from any highly correlated measures ($r \geq 0.50$) obtained under the first objective, and that the NEIS components have low to medium, rather than high correlations with theoretically less relevant constructs. Regarding the latter objective, because NEIS captures people's need for different input components to attain positive or reduce negative feelings, I probed the relationship between NEIS and constructs that assess people's general propensity to various affective states (e.g. need for affect, [62]). Finally, I aimed to explore the link between NEIS and other core personality traits, including BIG 5 [63].

## 6.1. Methods section

### 6.1.1. Participants, procedure and measures

Participants included in analyses are reported in table 2, whereas the information about all participants and the exclusion criteria are detailed in the electronic supplementary material, (pp. 4–5). Sample size was determined via *a priori* power analyses (for details, see electronic supplementary material, p. 22) computed to detect a medium correlation effect size ($r = 0.30$), assuming the power of 0.95. The analyses accounted for the false discovery rate (FDR) corrections [64] given multiple significance tests.

At the start, participants first completed the consent form, after which they responded to NEIS and the remaining personality measures. All measures are listed in table 4, alongside their descriptive statistics. Consistent with the introduction, the measures are divided into those assessing the needs for external input, people's inclinations and propensity to various affective states, and general personality.

**Table 4.** Measures used in Study 2 and their descriptive statistics. *Note*. NEIS = need for external input scale; NEIS-S = NEIS social input; NEIS-M = NEIS material input; NEIS-SS = NEIS sensation seeking input; IOS = interpersonal orientation scale [4]; FSMI = fundamental social motives inventory [8]; HVICS = horizontal–vertical individualism-collectivism scale [12]; MVS = material values scale, 15 item version [5]; DGS = dispositional greed scale [17]; BIS = buying impulsiveness scale [18]; ZKPQ-50-CC = cross-cultural shortened form of Zuckerman–Kuhlman personality inventory [20]; AISS = Arnett inventory of sensation seeking [19]; BPS-SR: boredom proneness scale—short form [65]; PANAS = positive and negative affect schedule [66]; NAQ = need for affect questionnaire [62]; MAS = mood awareness scale [67]; SAIS = short affect intensity scale [68]; SWLS = satisfaction with life scale [69]; TIPI = ten item personality inventory [63]; AATS = approach avoidance temperament scale [70]. All the scales, except for PANAS, were assessed on a 7-point Likert scale from 'Strongly disagree' to 'Strongly agree'. PANAS was assessed on a 5-point Likert scale from 'Very slightly or not at all' to 'Extremely'.

| measure | N | min | max | M | s.d. |
|---|---|---|---|---|---|
| NEIS | | | | | |
| NEIS-S | 317 | 1 | 7 | 3.211 | 1.393 |
| NEIS-M | 317 | 1 | 7 | 2.895 | 1.433 |
| NEIS-SS | 317 | 1 | 7 | 3.650 | 1.413 |
| external input scales | | | | | |
| social input | | | | | |
| IOS: emotional support | 317 | 1 | 7 | 3.905 | 1.636 |
| IOS: positive stimulation | 317 | 1 | 7 | 4.178 | 1.384 |
| FSMI: affiliation (group) | 317 | 1 | 7 | 4.500 | 1.302 |
| FSMI: affiliation (exclusion concern) | 317 | 1 | 7 | 3.930 | 1.624 |
| FSMI: affiliation (independence) | 317 | 1 | 7 | 5.084 | 1.292 |
| HVICS: horizontal individualism | 317 | 1 | 7 | 5.733 | 0.972 |
| HVICS: horizontal collectivism | 317 | 1 | 7 | 4.978 | 1.212 |
| HVICS: vertical individualism | 317 | 1 | 7 | 3.845 | 1.383 |
| HVICS: vertical collectivism | 317 | 1 | 7 | 4.932 | 1.257 |
| material input | | | | | |
| MVS: success | 317 | 1 | 7 | 3.250 | 1.475 |
| MVS: centrality | 317 | 1 | 7 | 3.444 | 1.275 |
| MVS: happiness | 317 | 1 | 7 | 4.036 | 1.459 |
| DGS: dispositional greed | 317 | 1 | 7 | 3.305 | 1.444 |
| BIS: buying impulsiveness | 317 | 1 | 7 | 2.776 | 1.404 |
| sensation seeking input | | | | | |
| ZKPQ-50-CC—impulsive sensation seeking | 317 | 1 | 7 | 3.326 | 1.405 |
| AISS: novelty | 317 | 1.6 | 6.4 | 4.174 | 0.899 |
| AISS: intensity | 317 | 1 | 7 | 3.677 | 1.011 |
| BPS-SR: boredom proneness | 317 | 1 | 7 | 2.923 | 1.158 |
| affect scales | | | | | |
| PANAS: positive | 317 | 1 | 5 | 3.016 | 0.836 |
| PANAS: negative | 317 | 1 | 4.2 | 1.559 | 0.663 |
| NAQ: approach | 317 | 1 | 7 | 4.670 | 1.192 |
| NAQ: avoidance | 317 | 1 | 7 | 3.146 | 1.442 |
| MAS: mood labelling | 317 | 1 | 7 | 5.203 | 1.313 |
| MAS: mood monitoring | 317 | 1 | 7 | 4.508 | 1.416 |
| SAIS: positive | 317 | 1 | 7 | 4.555 | 1.387 |
| SAIS: negative | 317 | 1 | 7 | 4.706 | 1.269 |

**Table 4.** (*Continued.*)

| measure | N | min | max | M | s.d. |
|---|---|---|---|---|---|
| SAIS: reverse positive | 317 | 1 | 7 | 3.279 | 1.277 |
| SWLS: life satisfaction | 317 | 1 | 7 | 4.305 | 1.705 |
| general personality | | | | | |
| TIPI: BIG 5 extraversion | 317 | 1 | 7 | 3.377 | 1.808 |
| TIPI: BIG 5 agreeableness | 317 | 1.5 | 7 | 5.415 | 1.328 |
| TIPI: BIG 5 conscientiousness | 317 | 2.5 | 7 | 5.678 | 1.123 |
| TIPI: BIG 5 openness | 317 | 1 | 7 | 5.153 | 1.405 |
| TIPI: BIG 5 emotional stability | 317 | 1 | 7 | 4.929 | 1.576 |
| AATS: approach temperament | 317 | 1 | 7 | 4.779 | 1.212 |
| AATS: avoidance temperament | 317 | 1 | 7 | 4.125 | 1.512 |

For clarity, the external input scales are further divided into those capturing social, material and sensation seeking input. All the measures were presented to participants in a randomized order. At the end, their demographics (age, gender and nationality) were assessed, and they were debriefed.

## 6.2. Results

Pearson correlations between the three NEIS components and the remaining constructs from Study 2 are presented in table 5. Regarding convergent validity, NEIS-S, NEIS-M and NEIS-SS were generally most strongly related to the measures capturing the corresponding input components, and correlation effect sizes ranged from medium to high (table 5). One exception important to discuss concerns vertical individualism, which had somewhat larger correlations with NEIS-M and NEIS-SS relative to NEIS-S. Because vertical individualism evaluates whether people like to compete and be better than others [12], it does not tap specifically into the need for social input. Its relatively larger correlations with NEIS-M and NEIS-SS suggest that people desire competing with others both for material reasons and to gain new and intense sensations.

As expected, NEIS-S, NEIS-M and NEISS-SS were also in most cases correlated with the scales capturing input components beyond the ones directly compatible with them, and these correlations ranged from small to medium (table 5). Study 2 therefore provides additional support for the notion that the three needs are linked and may belong to an overarching construct of external input.

Concerning discriminant validity, the correlations between NEIS subscales and affect-related variables were generally small; out of the 30 correlation coefficients computed, only four were of medium size, and none of them were large (table 5). These findings indicate that the needs for external input, despite being assessed via items that refer to positive and negative feelings, are distinct from scales that measure specifically affective experiences, thus establishing one aspect of discriminant validity of NEIS. As another aspect of discriminant validity, I also demonstrated that the three NEIS components are distinct from the theoretically relevant measures that were highly correlated with them (i.e. measures from table 5 that yielded correlations ≥ 0.50) using a series of factor analyses (see electronic supplementary material, p. 23)

Finally, the results showed that NEIS mostly had low correlations with general personality measures, with the exceptions being medium correlations between NEIS-S and extraversion or approach temperament, and between NEIS-M and approach temperament (table 5).

## 6.3. Discussion

Study 2 generated several key insights. First, I established convergent validity of NEIS by showing that its components had medium to high correlations with the scales capturing the corresponding forms of input. However, it was also observed that the NEIS needs were generally related with other non-identical forms of input (e.g. NEIS-S and material input). These findings provide additional insights regarding RQ1 because they document that each NEIS need is linked to all forms of input tackled in the present research (i.e. social, material and sensation seeking), regardless of how these are

**Table 5.** Pearson correlations between the NEIS components and the variables measured in Study 2. *Note.* $^*p < 0.05$, $^{**}p < 0.01$, $^{***}p < 0.001$. Raw significance values are reported: all the significant findings remained significant after the FDR correction [64]) was applied. NEIS = need for external input scale; NEIS-S = NEIS social input; NEIS-M = NEIS material input; NEIS-SS = NEIS sensation seeking input; IOS = interpersonal orientation scale; FSMI = fundamental social motives inventory; HVICS = horizontal–vertical individualism-collectivism scale; MVS = material values scale, 15 Item version; DGS = dispositional greed scale; BIS = buying impulsiveness scale; ZKPQ-50-CC = cross-cultural shortened form of Zuckerman–Kuhlman personality inventory; AISS = Arnett inventory of sensation seeking; BPS-SR: boredom proneness scale—short form; PANAS = positive and negative affect schedule; NAQ = need for affect questionnaire; MAS = mood awareness scale; SAIS = short affect intensity scale; SWLS = satisfaction with life scale; TIPI = ten item personality inventory; AATS = approach avoidance temperament scale.

| measure | NEIS | | |
| --- | --- | --- | --- |
| | NEIS-S | NEIS-M | NEIS-SS |
| NEIS | | | |
| NEIS-S | — | — | — |
| NEIS-M | 0.440*** | — | — |
| NEIS-SS | 0.537*** | 0.394*** | — |
| external input scales | | | |
| social input | | | |
| IOS: emotional support | 0.577*** | 0.194*** | 0.351*** |
| IOS: positive stimulation | 0.634*** | 0.203*** | 0.401*** |
| FSMI: affiliation (group) | 0.501*** | 0.101 | 0.279*** |
| FSMI: affiliation (exclusion concern) | 0.455*** | 0.461*** | 0.363*** |
| FSMI: affiliation (independence) | −0.577*** | −0.205*** | −0.283*** |
| HVICS: horizontal individualism | −0.263*** | −0.152** | −0.054 |
| HVICS: horizontal collectivism | 0.471*** | 0.012 | 0.265*** |
| HVICS: vertical individualism | 0.307*** | 0.423*** | 0.325*** |
| HVICS: vertical collectivism | 0.292*** | 0.104 | 0.137* |
| material input | | | |
| MVS: success | 0.348*** | 0.716*** | 0.228*** |
| MVS: centrality | 0.324*** | 0.711*** | 0.227*** |
| MVS: happiness | 0.191*** | 0.627*** | 0.203*** |
| DGS: dispositional greed | 0.279*** | 0.685*** | 0.303*** |
| BIS: buying impulsiveness | 0.301*** | 0.544*** | 0.364*** |
| sensation seeking input | | | |
| ZKPQ-50-CC—impulsive sensation seeking | 0.381*** | 0.218*** | 0.611*** |
| AISS: novelty | 0.231*** | 0.047 | 0.436*** |
| AISS: intensity | 0.147** | 0.158** | 0.345*** |
| BPS-SR: boredom proneness | 0.242*** | 0.369*** | 0.360*** |
| affect scales | | | |
| PANAS: positive | 0.310*** | 0.035 | 0.133* |
| PANAS: negative | 0.066 | 0.186*** | 0.163** |
| NAQ: approach | 0.369*** | 0.015 | 0.230*** |
| NAQ: avoidance | 0.049 | 0.243*** | 0.145** |
| MAS: mood labelling | −0.091 | −0.164** | −0.127* |
| MAS: mood monitoring | 0.281*** | 0.092 | 0.291*** |
| SAIS: positive | 0.436*** | 0.204*** | 0.361*** |

(*Continued.*)

| measure | NEIS | | |
| --- | --- | --- | --- |
| | NEIS-S | NEIS-M | NEIS-SS |
| SAIS: negative | 0.206*** | 0.158** | 0.168** |
| SAIS: reverse positive | 0.236*** | 0.160** | 0.225*** |
| SWLS: life satisfaction | 0.228*** | −0.067 | −0.020 |
| general personality | | | |
| TIPI: BIG 5 extraversion | 0.335*** | 0.091 | 0.184*** |
| TIPI: BIG 5 agreeableness | 0.226*** | −0.103 | 0.018 |
| TIPI: BIG 5 conscientiousness | −0.026 | −0.194*** | −0.110 |
| TIPI: BIG 5 openness | 0.095 | −0.177** | 0.257*** |
| TIPI: BIG 5 emotional stability | 0.020 | −0.142* | −0.101 |
| AATS: approach temperament | 0.433*** | 0.190*** | 0.402*** |
| AATS: avoidance temperament | 0.074 | 0.182** | 0.120* |

measured. Second, the present study established discriminant validity of NEIS by showing that (i) the needs it measures mostly had low correlations with the affect scales, and (ii) these needs were distinct from other scales tapping into external input that were highly correlated with them ($r \geq 0.50$). Third, Study 2 showed that the general personality construct most closely associated with the needs for input is approach temperament, given that it was linked to all three needs and had medium correlations with NEIS-S and NEIS-SS. Moreover, NEIS-S had a medium correlation with extraversion, which is consistent with the notion that some elements of this trait (e.g. being outgoing and sociable; [71]) are indicative of social input.

## 7. Study 3: long-term input deprivation

Study 3 addressed RQ2 by investigating the link between the needs for input and psychological experiences during a naturally occurring circumstance where external input was reduced long-term: COVID-19 pandemic. Namely, during the first wave of restrictions in 2020, I assessed the three needs via NEIS (part 1 of the study) and then contacted participants after 10 days to evaluate their emotional experiences and several behaviours (part 2). The emotion variables included positive experiences such as meaningfulness and positive feelings, and negative experiences such as anxiety and negative feelings (table 6). Moreover, the assessed behaviours included those that were strongly discouraged during lockdowns but that could increase people's exposure to external input, such as leaving the house for non-essential activities (table 6).

For each of these dependent variables, NEIS-S, NEIS-M and NEIS-SS were examined as predictors in the same regression models. Although I did not have clear hypotheses given the lack of previous research, I had some general expectations. I assumed that COVID-19 restrictions may be more detrimental to social and sensation seeking input than to material input because people were discouraged from leaving the house and could therefore not easily socialize or engage in novel and stimulating activities, but they could still order material objects (via online shopping). Therefore, I expected NEIS-S and NEIS-SS may be more likely to significantly predict the dependent variables than NEIS-M. Moreover, because COVID-19 restrictions were enforced by penalties in many of the tested countries, I expected NEIS subscales may be less predictive of behaviours than experiences.

Finally, in Study 3, I assessed incremental predictive validity by probing whether the NEIS needs would predict how much people required the corresponding type of input (social, material and sensation seeking) during the 10 days of COVID-19 restrictions beyond and above several competing scales that yielded strong correlations ($r \geq 0.50$) in Study 2 (table 6). An additional study that examined incremental predictive validity of NEIS against a broader range of scales is available in the electronic supplementary material (pp. 24–37).

**Table 6.** Main measures used in Study 3 and their descriptive statistics. *Note.* NEIS = need for external input scale; NEIS-S = NEIS social input; NEIS-M = NEIS material input; NEIS-SS = NEIS sensation seeking input; IOS = interpersonal orientation scale [4]; MVS = material values scale, 15 item version [5]; ZKPQ-50-CC = cross-cultural shortened form of Zuckerman–Kuhlman personality inventory [20]. NEIS, all external input scales and all incremental predictive validity variables were assessed on a 7-point Likert scale from 'Strongly disagree' to 'Strongly agree'. For meaningfulness, anxiety, life satisfaction, relative life satisfaction and social distancing, a 5-point Likert scale from 'Not at all' to 'Extremely' was used. Positive and negative feelings were assessed on a 5-point Likert scale from 'Very rarely or never' to 'Very often or always', and depression was assessed on a 4-point Likert Scale from 'Not at all' to 'Nearly every day'. Finally, all 'behaviour'-dependent variables, except for social distancing, were assessed on a scale from '0 = I stayed at home all the time/Never' to '36 = more than 35 times'.

| measure | N | min | max | M | s.d. |
|---|---|---|---|---|---|
| NEIS | | | | | |
| NEIS-S | 1992 | 1 | 7 | 3.766 | 1.373 |
| NEIS-M | 1992 | 1 | 6.5 | 2.387 | 1.022 |
| NEIS-SS | 1992 | 1 | 7 | 4.093 | 1.346 |
| external input scales | | | | | |
| IOS: emotional support | 1992 | 1 | 7 | 4.227 | 1.380 |
| IOS: positive stimulation | 1992 | 1 | 7 | 4.548 | 1.104 |
| MVS: success | 1992 | 1 | 6.2 | 2.317 | 1.001 |
| MVS: centrality | 1992 | 1 | 6.8 | 3.388 | 0.956 |
| MVS: happiness | 1992 | 1 | 6.8 | 2.875 | 1.187 |
| ZKPQ-50-CC—impulsive sensation seeking | 1992 | 1 | 7 | 3.740 | 1.156 |
| dependent variables: emotional experiences | | | | | |
| meaningfulness (How meaningful participants found the 10-day period during COVID-19 restrictions; [72]) | 1992 | 1 | 5 | 3.455 | 1.046 |
| anxiety (How anxious participants felt during the 10-day period; [73]) | 1992 | 1 | 5 | 2.610 | 0.974 |
| positive feelings (Positive feelings experienced during the 10-day period; [74]) | 1992 | 1 | 5 | 3.264 | 0.743 |
| negative feelings (Negative feelings experienced during the 10-day period; [74]) | 1992 | 1 | 5 | 2.538 | 0.764 |
| depression (Depressive symptoms during the 10-day period; [75]) | 1992 | 0 | 27 | 6.694 | 5.418 |
| life satisfaction post (Life satisfaction measured at the end of the 10-day period; [76]) | 1991 | 1 | 5 | 3.236 | 0.967 |
| relative life satisfaction (Difference between life satisfaction measured at the end versus beginning of the 10-day period) | 1991 | −4 | 3 | −0.089 | 0.879 |
| dependent variables: behaviour | | | | | |
| social distancing (General compliance with social distancing behaviours such as working from home whenever possible or not leaving the house except for essential activities over the 10-day period; [77]) | 1992 | 1 | 5 | 4.811 | 0.459 |
| leaving the house (How many times during the 10-day period participants left the house for any activities except for the essential ones, including buying medication, going to the doctor, buying food and working if 'essential worker'; [77]) | 1985 | 0 | 36 | 3.375 | 4.507 |

(*Continued.*)

**Table 6.** (*Continued.*)

| measure | N | min | max | M | s.d. |
|---|---|---|---|---|---|
| leaving the house to buy groceries (How many times during the 10-day period participants left the house to do grocery shopping) | 1991 | 0 | 20 | 1.879 | 1.818 |
| meeting others (How many times during the 10-day period participants left the house to socialize with other or had someone visit them) | 1990 | 0 | 10 | 0.224 | 0.878 |
| dependent variables: incremental predictive validity | | | | | |
| social input (Assessed via four items, $\alpha = 0.648$, that probed how much people missed face-to-face interaction, missed going out of their house to be around others, engaged in virtual social interaction, and kept in touch with their close ones who live elsewhere over the 10-day period) | 1991 | 1.25 | 7 | 5.762 | 0.996 |
| material input (Assessed via four items, $\alpha = 0.741$, that probed how much participants shopped online [excluding food], browsed online stores and products [excluding food], missed browsing physical products, and missed shopping in physical stores [excluding food] over the 10-day period) | 1991 | 1 | 7 | 3.196 | 1.510 |
| sensation seeking Input 1 (Assessed via two items, $r = 0.360$, that probed whether people sought ways to stimulate themselves to make their life interesting, and tried many unusual things to amuse themselves over the 10-day period)[a] | 1991 | 1 | 7 | 4.437 | 1.200 |
| sensation seeking Input 2 (Assessed via two items, $r = 0.681$, that probed whether participants felt that their life was monotonous and felt deprived of input over the 10-day period)[a] | 1991 | 1 | 7 | 4.032 | 1.713 |

[a]Although I was initially planning to combine items measuring sensation seeking Input 1 and 2 into a single sensation seeking input variable, their composite score had a low Cronbach's $\alpha$ (0.370), and some items were negatively correlated. I therefore split the items into two variables based on their thematic similarity and correlations between the items.

## 7.1. Methods section

### 7.1.1. Participants and procedure

Participants included in analyses are reported in table 2, whereas the information about all participants and the exclusion criteria are detailed in the electronic supplementary material (pp. 4–5). Because pandemics are relatively rare real-life circumstances, I aimed to maximize the sample size rather than meet a specific target driven by *a priori* power analysis. Sensitivity power analyses [78] that were based on participants who were included in statistical analyses and accounted for the FDR corrections [64] applied to significance tests showed that the study was highly powered (i.e. 0.95) to capture small effects (Cohen's $f^2 = 0.016$) concerning the link between NEIS needs and the dependent variables (electronic supplementary material, pp. 38–39).

In part 1, participants first completed the consent form, after which demographics (nationality and country of residence) and covariates were assessed. In that context, it is important to emphasize that the study did not focus on a particular type of participants (e.g. those who lived alone or were used to spending time at home). Instead, various variables relevant to psychological experiences and compliance regarding COVID-19 were measured as covariates and controlled for in statistical analyses

to avoid potential confounding effects (see the Results section below). Examples of covariates include people's distancing history (i.e. when they first started practising social distancing), household size (i.e. how many people lived together with the participant in the same household), living situation (i.e. whether participants' living situation allowed them to comply with social distancing) or being used to spending time at home (i.e. how many full days participants would typically spend at home before the COVID-19 pandemic started). A comprehensive list and description of all the covariates measured is available in the electronic supplementary material (pp. 42–43).

In addition to the demographics and covariates, in part 1, participants completed NEIS and the measures assessing their baseline life satisfaction [76] and external input (table 6); these measures were presented in a randomized order. Participants were then contacted after 10 days for part 2. They again completed the consent form and subsequently answered the questions measuring the dependent variables: emotional experiences, behaviour and incremental predictive validity (table 6). In the end, participants were debriefed.

## 7.2. Results

To investigate which NEIS subscales would predict the dependent variables (table 6), 15 multiple linear regressions were computed. Table 7 presents these analyses in their abbreviated format (i.e. without covariates, which were used as control variables in all analyses) due to the substantial amount of output; full analysis output is available in the electronic supplementary material (pp. 44–60). In models 12–15 (table 7), the competing external input scales were included alongside NEIS as predictors because these analyses examined incremental predictive validity of NEIS. Pearson correlations between NEIS and all dependent variables are available in the electronic supplementary material (p. 61).

As table 7 shows, higher scores on NEIS-S and NEIS-SS were generally associated with more negative emotional experiences (apart from relative life satisfaction), whereas NEIS-M did not significantly predict any of the experiences. Concerning behaviour, NEIS-SS was the key predictor: higher scores on this variable were associated with leaving the house more frequently. The scale generally did not predict social distancing and meeting others. Finally, because NEIS subscales significantly predicted the corresponding input variables beyond the competing scales, its incremental predictive validity was demonstrated. Although for *sensation seeking input 2* NEIS-S yielded a larger effect size than NEIS-SS, only the latter subscale consistently predicted each *sensation seeking input* variable.

## 7.3. Discussion

To address RQ2, Study 3 demonstrated that the needs for social and sensation seeking input, but not the need for material input, were generally negatively associated with people's experiences of long-term input deprivation created by COVID-19 restrictions. This is consistent with my speculation that the restrictions may be more detrimental to people's access to social and sensation seeking input versus material input, in which case the need for material input would be less relevant to people's emotional experiences during this period. Although I speculated that personality might be more likely to predict emotional experiences than COVID-19 compliance behaviours, which were typically regulated by external factors beyond personality (e.g. they were enforced by fines), NEIS did predict several behaviours. More precisely, people higher on NEIS-SS were more likely to leave the house to pursue non-essential activities and to buy groceries. Finally, next to revealing the importance of the need for external input for people's experiences of and behaviour during long-term input deprivation, Study 3 demonstrated incremental predictive validity of NEIS, given that its subscales predicted the corresponding input variables beyond the competing scales.

# 8. Study 4: short-term input deprivation

Study 4 addressed RQ2 by examining the link between the needs for input and psychological experiences and behaviour during short-term input deprivation—a brief 12 min period, referred to as a free-time period, during which participants were asked to do nothing and entertain themselves with their thoughts (see [37,80]). Regarding psychological experiences (table 8), I examined positive feelings such as pleasantness or meaningfulness during the free-time period and negative experiences such as boredom, intrusive thoughts and difficulty to focus [37,82,84]. I also probed several experiences relevant to different input components (table 8): whether participants felt under-stimulated and had

**Table 7.** Multiple linear regressions regarding the link between NEIS and the dependent variables concerning experiences, behaviour and incremental predictive validity (Study 3). *Note.* Model 1 $R^2 = 0.193$; Model 2 $R^2 = 0.199$; Model 3 $R^2 = 0.207$; Model 4 $R^2 = 0.287$; Model 5 $R^2 = 0.299$; Model 6 $R^2 = 0.205$; Model 7 $R^2 = 0.019$; Model 8 $R^2 = 0.061$; Model 9 $R^2 = 0.078$; Model 10 $R^2 = 0.078$; Model 11 $R^2 = 0.049$; Model 12 $R^2 = 0.317$; Model 13 $R^2 = 0.210$; Model 14 $R^2 = 0.090$; Model 15 $R^2 = 0.278$. Out of 1992 participants who were included in statistical analyses, in Models 1–5, 8, 10, and 12–15, 1959 participants were analysed due to missing data; in Models 6, 7 and 11, 1958 participants were analysed due to missing data; and in Model 9, 1957 participants were analysed due to missing data. IOS = interpersonal orientation scale; MVS = material values scale; ZKPQ-50-CC = cross-cultural shortened form of Zuckerman–Kuhlman personality inventory. Given the substantial output of the analyses, only the key predictors are displayed in table 7, whereas full regression models that include the covariates as predictors are available in the electronic supplementary material, pp. 44–60. $f^2$ denotes Cohen's $f^2$ effect size [79]: effects ≤ 0.02 are considered small. Raw significance values are reported: symbol † indicates results that stopped being significant after applying the FDR correction [64]. The correction was applied across both the key predictors and covariates.

| predictor | b | s.e. b | 95% CI | t | p | $f^2$ |
|---|---|---|---|---|---|---|
| EXPERIENCES | | | | | | |
| Model 1 (DV = meaningfulness) | | | | | | |
| (intercept) | 1.198 | 0.480 | 0.257–2.139 | 2.496 | 0.013 | 0.003 |
| NEIS-S | −0.068 | 0.019 | −0.106 to −0.030 | −3.503 | <0.001 | 0.006 |
| NEIS-M | −0.006 | 0.023 | −0.051–0.039 | −0.278 | 0.781 | <0.001 |
| NEIS-SS | −0.110 | 0.019 | −0.149 to −0.072 | −5.672 | <0.001 | 0.017 |
| Model 2 (DV = anxiety) | | | | | | |
| (intercept) | 3.407 | 0.445 | 2.535–4.279 | 7.662 | <0.001 | 0.030 |
| NEIS-S | 0.078 | 0.018 | 0.043–0.113 | 4.357 | <0.001 | 0.010 |
| NEIS-M | 0.017 | 0.021 | −0.024–0.059 | 0.821 | 0.412 | <0.001 |
| NEIS-SS | 0.017 | 0.018 | −0.018–0.053 | 0.966 | 0.334 | <0.001 |
| Model 3 (DV = positive experiences) | | | | | | |
| (intercept) | 1.774 | 0.338 | 1.111–2.438 | 5.244 | <0.001 | 0.014 |
| NEIS-S | −0.087 | 0.014 | −0.114 to −0.060 | −6.378 | <0.001 | 0.021 |
| NEIS-M | −0.007 | 0.016 | −0.039–0.025 | −0.438 | 0.662 | <0.001 |
| NEIS-SS | −0.058 | 0.014 | −0.085 to −0.031 | −4.231 | <0.001 | 0.009 |
| Model 4 (DV = negative experiences) | | | | | | |
| (intercept) | 3.668 | 0.329 | 3.022–4.314 | 11.135 | <0.001 | 0.064 |
| NEIS-S | 0.078 | 0.013 | 0.052–0.104 | 5.856 | <0.001 | 0.018 |
| NEIS-M | 0.026 | 0.016 | −0.005–0.057 | 1.666 | 0.096 | 0.001 |
| NEIS-SS | 0.047 | 0.013 | 0.020–0.073 | 3.486 | 0.001 | 0.006 |
| Model 5 (DV = depression) | | | | | | |
| (intercept) | 15.228 | 2.320 | 10.678–19.778 | 6.564 | <0.001 | 0.022 |
| NEIS-S | 0.401 | 0.093 | 0.217–0.584 | 4.285 | <0.001 | 0.010 |
| NEIS-M | 0.178 | 0.111 | −0.040–0.395 | 1.598 | 0.110 | 0.001 |
| NEIS-SS | 0.452 | 0.094 | 0.268–0.637 | 4.804 | <0.001 | 0.012 |
| Model 6 (DV = life satisfaction post) | | | | | | |
| (intercept) | 1.546 | 0.440 | 0.683–2.409 | 3.513 | <0.001 | 0.006 |

(*Continued.*)

| predictor | *b* | s.e. *b* | 95% CI | *t* | *p* | *f*² |
|---|---|---|---|---|---|---|
| NEIS-S | −0.107 | 0.018 | −0.142 to −0.072 | −6.014 | <0.001 | 0.019 |
| NEIS-M | −0.031 | 0.021 | −0.072–0.011 | −1.451 | 0.147 | 0.001 |
| NEIS-SS | −0.062 | 0.018 | −0.097 to −0.027 | −3.495 | <0.001 | 0.006 |
| Model 7 (DV = relative life satisfaction) | | | | | | |
| (intercept) | −0.238 | 0.444 | −1.109–0.632 | −0.537 | 0.591 | <0.001 |
| NEIS-S | 0.017 | 0.018 | −0.018–0.052 | 0.929 | 0.353 | <0.001 |
| NEIS-M | −0.021 | 0.021 | −0.063–0.020 | −0.998 | 0.319 | 0.001 |
| NEIS-SS | 0.015 | 0.018 | −0.021–0.050 | 0.812 | 0.417 | <0.001 |
| BEHAVIOUR | | | | | | |
| Model 8 (DV = social distancing) | | | | | | |
| (intercept) | 4.341 | 0.227 | 3.897–4.785 | 19.160 | <0.001 | 0.191 |
| NEIS-S | 0.011 | 0.009 | −0.007–0.029 | 1.199 | 0.231 | 0.001 |
| NEIS-M | 0.022 | 0.011 | 0.001–0.043 | 2.016 | 0.044† | 0.002 |
| NEIS-SS | −0.019 | 0.009 | −0.037 to −0.001 | −2.065 | 0.039† | 0.002 |
| Model 9 (DV = leaving the house) | | | | | | |
| (intercept) | 1.964 | 2.183 | −0.318–6.246 | 0.900 | 0.368 | <0.001 |
| NEIS-S | −0.027 | 0.088 | −0.200–0.145 | −0.313 | 0.755 | <0.001 |
| NEIS-M | −0.070 | 0.105 | −0.275–0.136 | −0.665 | 0.506 | <0.001 |
| NEIS-SS | 0.363 | 0.089 | 0.189–0.537 | 4.099 | <0.001 | 0.009 |
| Model 10 (DV = leaving the house to buy groceries) | | | | | | |
| (intercept) | 2.278 | 0.865 | 0.582–3.974 | 2.634 | 0.009 | 0.004 |
| NEIS-S | 0.023 | 0.035 | −0.045–0.092 | 0.673 | 0.501 | <0.001 |
| NEIS-M | 0.027 | 0.041 | −0.055–0.108 | 0.640 | 0.522 | <0.001 |
| NEIS-SS | 0.099 | 0.035 | 0.031–0.168 | 2.835 | 0.005 | 0.004 |
| Model 11 (DV = meeting others) | | | | | | |
| (intercept) | 0.177 | 0.440 | −0.686–1.040 | 0.403 | 0.687 | <0.001 |
| NEIS-S | 0.036 | 0.018 | 0.002–0.071 | 2.053 | 0.040† | 0.002 |
| NEIS-M | 0.013 | 0.021 | −0.028–0.055 | 0.631 | 0.528 | <0.001 |
| NEIS-SS | 0.025 | 0.018 | −0.010–0.060 | 1.376 | 0.169 | 0.001 |
| incremental predictive validity | | | | | | |
| Model 12 (DV = social input) | | | | | | |
| (intercept) | 4.012 | 0.423 | 3.183–4.842 | 9.485 | <0.001 | 0.047 |
| NEIS-S | 0.130 | 0.021 | 0.089–0.171 | 6.241 | <0.001 | 0.020 |
| NEIS-M | 0.000 | 0.020 | −0.039–0.040 | 0.024 | 0.981 | <0.001 |
| NEIS-SS | 0.042 | 0.017 | 0.008–0.075 | 2.430 | 0.015† | 0.003 |
| IOS: emotional support | 0.153 | 0.020 | 0.114–0.191 | 7.715 | <0.001 | 0.031 |
| IOS: positive stimulation | 0.126 | 0.023 | 0.080–0.172 | 5.391 | <0.001 | 0.015 |
| Model 13 (DV = material input) | | | | | | |
| (intercept) | 0.963 | 0.699 | −0.408–2.334 | 1.377 | 0.169 | 0.001 |

(*Continued.*)

| predictor | *b* | s.e. *b* | 95% CI | *t* | *p* | *f²* |
|---|---|---|---|---|---|---|
| NEIS-S | 0.072 | 0.028 | 0.018–0.127 | 2.629 | 0.009 | 0.004 |
| NEIS-M | 0.297 | 0.041 | 0.216–0.378 | 7.216 | <0.001 | 0.027 |
| NEIS-SS | 0.039 | 0.028 | −0.016–0.094 | 1.403 | 0.161 | 0.001 |
| MVS: success | −0.019 | 0.040 | −0.098–0.059 | −0.483 | 0.629 | <0.001 |
| MVS: centrality | 0.311 | 0.042 | 0.227–0.394 | 7.314 | <0.001 | 0.028 |
| MVS: happiness | 0.054 | 0.034 | −0.013–0.121 | 1.576 | 0.115 | 0.001 |
| Model 14 (DV = sensation seeking Input 1) | | | | | | |
| (intercept) | 1.850 | 0.588 | 0.697–3.002 | 3.147 | 0.002 | 0.005 |
| NEIS-S | −0.049 | 0.024 | −0.095 to −0.002 | −2.058 | 0.040† | 0.002 |
| NEIS-M | −0.004 | 0.028 | −0.059–0.051 | −0.153 | 0.878 | <0.001 |
| NEIS-SS | 0.093 | 0.025 | 0.043–0.142 | 3.667 | <0.001 | 0.007 |
| ZKPQ-50-CC: impulsive sensation seeking | 0.050 | 0.030 | −0.008–0.108 | 1.678 | 0.093 | 0.001 |
| Model 15 (DV = sensation seeking Input 2) | | | | | | |
| (intercept) | 4.815 | 0.747 | 3.350–6.280 | 6.445 | <0.001 | 0.022 |
| NEIS-S | 0.277 | 0.030 | 0.218–0.336 | 9.222 | <0.001 | 0.044 |
| NEIS-M | 0.112 | 0.036 | 0.042–0.182 | 3.143 | 0.002 | 0.005 |
| NEIS-SS | 0.228 | 0.032 | 0.165–0.291 | 7.088 | <0.001 | 0.026 |
| ZKPQ-50-CC: impulsive sensation seeking | 0.100 | 0.038 | 0.026–0.174 | 2.652 | 0.008 | 0.004 |

the urge to communicate with someone, interact with material objects and use technology. Finally, arousal was measured because I wanted to understand whether heightened need for input is linked to lower arousal levels, which might indicate that some people need more input to maintain optimal arousal levels (see [85]). Regarding behaviours, I focused on cheating. Buttrick *et al.* [37] showed that 54% of participants cheated by engaging in activities they were asked to avoid during the free-time period (e.g. checking e-mail and listening to music). I therefore assumed that establishing a link between the needs for external input and cheating would be a powerful demonstration of how much these needs shape people's propensity to withstand short-term input deprivation.

For all dependent variables (i.e. experiences and behaviours), I examined NEIS-S, NEIS-M and NEIS-SS as predictors in the same regression models. Given the lack of previous research, I did not have clear hypotheses. Based on the evidence linking enjoyment during short-term input deprivation and daily smartphone use as a potential indicator of materialism [37], it was possible to speculate only that the need for material input might be associated with participants' experiences of the free-time period.

## 8.1. Methods section

### 8.1.1. Participants and procedure

Participants included in analyses are reported in table 2, whereas the information about all participants and the exclusion criteria are detailed in the electronic supplementary material (pp. 4–5). Sample size was determined via *a priori* power analyses (for details, see electronic supplementary material, pp. 62–63) computed to detect Cohen's *f²* of 0.065 (a midpoint between medium—0.15—and small—0.02—effects; [79]) concerning the link between a NEIS component and a dependent variable in a multiple regression (power = 0.95). The power analyses accounted for the FDR corrections [64], given multiple significance tests.

**Table 8.** Main measures used in Study 4 and their descriptive statistics. *Note.* NEIS = need for external input scale; NEIS-S = NEIS social input; NEIS-M = NEIS material input; NEIS-SS = NEIS sensation seeking input. NEIS was assessed on a 7-point Likert scale from 'Strongly disagree' to 'Strongly agree'. All experiences, except for arousal, were measured using a 9-point scale from '1 = Not at all' to '9 = Very much/Extremely'. Arousal was measured using a 9-point self-assessment manikin ranging from low to high arousal. Cheating was measured on a scale from '0 = 0 min (throughout the entire period, I did nothing except for trying to entertain myself with my thoughts' to '12 = Up to 12 min)'.

| measure | N | min | max | M | s.d. |
|---|---|---|---|---|---|
| NEIS | | | | | |
| NEIS-S | 519 | 1 | 7 | 3.990 | 1.690 |
| NEIS-M | 519 | 1 | 7 | 4.001 | 1.684 |
| NEIS-SS | 519 | 1 | 7 | 4.434 | 1.391 |
| dependent variables: experiences | | | | | |
| pleasantness (Pleasantness experienced during the free-time period; [37])[a] | 519 | 1 | 9 | 5.820 | 2.278 |
| meaningfulness (How meaningful participants found the free-time period; [72]) | 519 | 1 | 9 | 6.060 | 2.312 |
| boredom (How boring participants found the free-time period, [37])[a] | 519 | 1 | 9 | 5.358 | 2.537 |
| intrusive thoughts (To what extent participants experienced, during the free-time period, unwanted mental images, ideas, or reflections they did not want to think about but that spontaneously kept recurring; [82]) | 519 | 1 | 9 | 4.911 | 2.644 |
| mind wandering (To what extent participants found their mind wandering during the free-time period; [37]) | 519 | 1 | 9 | 6.805 | 1.908 |
| hard to concentrate (How hard it was for participants to concentrate during the free-time period; [37]) | 518 | 1 | 9 | 5.556 | 2.308 |
| difficulty ideas (How difficult it was for participants to find ideas to think about during the free-time period) | 519 | 1 | 9 | 4.543 | 2.662 |
| urge to communicate (To what extent participants had the urge to communicate to someone during the free-time period) | 519 | 1 | 9 | 4.426 | 2.903 |
| urge for material objects (To what extent participants had the urge to interact with any material objects during the free-time period) | 519 | 1 | 9 | 5.453 | 2.547 |
| urge for technology (To what extent participants had the urge to check their smartphone or use some other technological device during the free-time period) | 519 | 1 | 9 | 5.156 | 2.866 |
| under-stimulated (To what extent participants felt under-stimulated during the free-time period) | 519 | 1 | 9 | 5.590 | 2.396 |
| arousal (Participants' level of arousal during the free-time period; [83]) | 519 | 1 | 9 | 5.073 | 2.034 |
| dependent variables: behaviour | | | | | |
| cheating (How much time in minutes participants spent on any other activities except for entertaining themselves with their thoughts— e.g. talking to someone, texting someone, using the phone, listening to music, checking e-mail, browsing the web, etc.) | 519 | 0 | 12 | 2.486 | 3.809 |

[a]Although Buttrick *et al.* [37] reverse-coded Boredom scores and combined this variable with the remaining positive affect items measuring Pleasantness, I analysed Boredom as a separate dependent variable because it is a distinct negative emotion whose absence does not conceptually correspond to pleasantness or enjoyability [81].

The study procedure closely followed Buttrick *et al.* [37]. All participants first completed the consent form and were then told the study would take 15–20 min of their time, and they would need to do it alone in their room. They were also told to put away any objects that may distract them and close all windows on their computer. After these initial instructions, the *free-time period* was introduced to participants. It was specified that the period would last 10–15 min (the actual duration was 12 min), and they were told to entertain themselves with their thoughts but avoid doing anything else. After participants completed the free-time period, they first received the questions measuring all dependent variables—experiences and the cheating behaviour (table 8)—after which they completed NEIS. Several covariates were also assessed (for details, see electronic supplementary material, pp. 65–66)—they included variables that were found to predict people's experiences during the free-time period, including meditation experience, smartphone use and need for cognition [37]. In the end, all participants were debriefed.

## 8.2. Results

To investigate which NEIS subscales would predict the dependent variables (table 8), 13 multiple linear regressions were computed. Table 9 presents these analyses in their abbreviated format (i.e. without covariates, which were used as control variables in all analyses); full analysis output is available in the electronic supplementary material (pp. 67–77). For Pearson correlations between NEIS and all dependent variables, see electronic supplementary material (p. 78).

As table 9 shows, NEIS-S was the only significant predictor of positive experiences (pleasantness and meaningfulness); unlike in Study 3, this subscale was positively linked to these experiences. Moreover, it positively predicted people's urge to communicate during the free-time period. Negative experiences— boredom and intrusive thoughts—were positively predicted only by NEIS-M. Higher scores on this subscale were also associated with more mind wandering, difficulty to concentrate and find ideas to think about, urge for material objects and technology, and feeling under-stimulated. Importantly, NEIS-M was the only subscale that positively predicted cheating. Finally, NEIS-SS predicted only one experience—the difficulty of finding ideas to think about. None of the NEIS subscales significantly predicted arousal.

## 8.3. Discussion

The present findings, in combination with Study 3, indicate that the needs for external input may differ regarding their temporal horizon. Whereas higher scores on NEIS-S were generally linked to more negative experiences of long-term input deprivation (Study 3), they were associated with higher pleasantness and meaningfulness under the short-term deprivation (Study 4). Moreover, even if NEIS-SS was negatively linked to wellbeing in Study 3, it was largely a non-significant predictor of experiences and behaviour in the present study. By contrast, whereas NEIS-M was generally a non-significant predictor in the previous study, it was the core predictor in Study 4, where it was associated with a greater difficulty of withstanding the free-time period. Namely, high NEIS-M scores were linked to more boredom, intrusive thoughts, mind wandering and cheating, increased difficulty to concentrate and find ideas to think about, feeling more under-stimulated, and a heightened urge for material objects more generally and technology more specifically.

# 9. General discussion

The present research offered preliminary insights into the psychology of external input by examining this construct in relation to the needs for material, social and sensation seeking input (RQ1), and by probing the link between these needs and psychological functioning during long- and short-term input deprivation (RQ2). Regarding RQ1, Study 1 showed that the three needs are related and constitute different dimensions of the need for external input as an overarching construct. Study 2 provided further insights relevant to RQ1 by showing that NEIS-S, NEIS-M and NEIS-SS were generally correlated with (i) the scales assessing the corresponding forms of input and (ii) the scales measuring the remaining forms of input (e.g. NEIS-S and sensation seeking input), thus substantiating the initial finding that different needs for input are interrelated.

Regarding RQ2, Study 3 showed that NEIS-S and NEIS-SS had negative implications for people's wellbeing during COVID-19 lockdowns, as they were negative predictors of positive emotional experiences and positive predictors of negative emotional experiences. Higher need for sensation seeking input also predicted higher likelihood of leaving the house to pursue non-essential activities. The need for material input was generally

**Table 9.** Multiple linear regressions regarding the link between NEIS and the dependent variables concerning experiences and behaviour (Study 4). *Note.* Model 1 $R^2 = 0.410$; Model 2 $R^2 = 0.313$; Model 3 $R^2 = 0.232$; Model 4 $R^2 = 0.392$; Model 5 $R^2 = 0.085$; Model 6 $R^2 = 0.266$; Model 7 $R^2 = 0.437$; Model 8 $R^2 = 0.465$; Model 9 $R^2 = 0.252$; Model 10 $R^2 = 0.285$; Model 11 $R^2 = 0.269$; Model 12 $R^2 = 0.191$; Model 13 $R^2 = 0.369$. Out of 519 participants who were included in statistical analyses, in Model 6, 514 participants were analysed due to missing data, whereas in the remaining models 515 participants were analysed due to missing data. Given the substantial output of the analyses, only the key predictors are displayed in table 9, whereas full regression models that include the covariates as predictors are available in the electronic supplementary material, pp. 67–77. $f^2$ denotes Cohen's $f^2$ effect size [79]: effects $\leq 0.02$ are considered small. Raw significance values are reported: symbol † indicates results that stopped being significant after applying the FDR correction [64]. The correction was applied across both the key predictors and covariates.

| predictor | $b$ | s.e. $b$ | 95% CI | $t$ | $p$ | $f^2$ |
|---|---|---|---|---|---|---|
| experiences | | | | | | |
| Model 1 (DV = pleasantness) | | | | | | |
| (intercept) | 0.985 | 0.751 | −0.491–2.461 | 1.311 | 0.190 | 0.003 |
| NEIS-S | 0.304 | 0.085 | 0.136–0.471 | 3.560 | <0.001 | 0.026 |
| NEIS-M | 0.035 | 0.083 | −0.129–0.199 | 0.420 | 0.675 | <0.001 |
| NEIS-SS | −0.149 | 0.093 | −0.332–0.033 | −1.608 | 0.109 | 0.005 |
| Model 2 (DV = meaningfulness) | | | | | | |
| (intercept) | 0.366 | 0.822 | −1.250–1.982 | 0.445 | 0.656 | <0.001 |
| NEIS-S | 0.282 | 0.093 | 0.099–0.466 | 3.021 | 0.003 | 0.018 |
| NEIS-M | −0.006 | 0.091 | −0.185–0.174 | −0.061 | 0.951 | <0.001 |
| NEIS-SS | −0.079 | 0.102 | −0.279–0.121 | −0.779 | 0.436 | 0.001 |
| Model 3 (DV = boredom) | | | | | | |
| (Intercept) | 7.804 | 0.955 | 5.927–9.681 | 8.169 | <0.001 | 0.135 |
| NEIS-S | −0.043 | 0.109 | −0.257–0.170 | −0.401 | 0.689 | <0.001 |
| NEIS-M | 0.355 | 0.106 | 0.147–0.563 | 3.348 | 0.001 | 0.023 |
| NEIS-SS | 0.179 | 0.118 | −0.053–0.411 | 1.517 | 0.130 | 0.005 |
| Model 4 (DV = intrusive thoughts) | | | | | | |
| (intercept) | 0.124 | 0.885 | −1.615–1.863 | 0.140 | 0.889 | <0.001 |
| NEIS-S | 0.193 | 0.101 | −0.004–0.391 | 1.924 | 0.055 | 0.007 |
| NEIS-M | 0.545 | 0.098 | 0.352–0.739 | 5.551 | <0.001 | 0.062 |
| NEIS-SS | 0.077 | 0.109 | −0.138–0.292 | 0.701 | 0.484 | 0.001 |
| Model 5 (DV = mind wandering) | | | | | | |
| (intercept) | 4.925 | 0.784 | 3.385–6.464 | 6.283 | <0.001 | 0.080 |
| NEIS-S | −0.109 | 0.089 | −0.284–0.066 | −1.227 | 0.220 | 0.003 |
| NEIS-M | 0.279 | 0.087 | 0.108–0.450 | 3.212 | 0.001 | 0.021 |
| NEIS-SS | 0.069 | 0.097 | −0.121–0.259 | 0.711 | 0.477 | 0.001 |
| Model 6 (DV = hard to concentrate) | | | | | | |
| (intercept) | 4.485 | 0.850 | 2.815–6.154 | 5.277 | <0.001 | 0.056 |
| NEIS-S | 0.033 | 0.097 | −0.157–0.222 | 0.337 | 0.736 | <0.001 |
| NEIS-M | 0.393 | 0.094 | 0.208–0.578 | 4.168 | <0.001 | 0.035 |
| NEIS-SS | 0.235 | 0.105 | 0.028–0.441 | 2.232 | 0.026† | 0.010 |
| Model 7 (DV = difficulty ideas) | | | | | | |
| (intercept) | 4.663 | 0.853 | 2.986–6.340 | 5.464 | <0.001 | 0.060 |
| NEIS-S | 0.134 | 0.097 | −0.056–0.325 | 1.383 | 0.167 | 0.004 |
| NEIS-M | 0.431 | 0.095 | 0.245–0.618 | 4.555 | <0.001 | 0.042 |

| predictor | $b$ | s.e. $b$ | 95% CI | $t$ | $p$ | $f^2$ |
|---|---|---|---|---|---|---|
| NEIS-SS | 0.319 | 0.106 | 0.112–0.527 | 3.026 | 0.003 | 0.019 |
| Model 8 (DV = urge to communicate) | | | | | | |
| (intercept) | −0.051 | 0.910 | −1.839–1.737 | −0.056 | 0.955 | <0.001 |
| NEIS-S | 0.334 | 0.103 | 0.131–0.537 | 3.229 | 0.001 | 0.021 |
| NEIS-M | 0.269 | 0.101 | 0.070–0.467 | 2.660 | 0.008† | 0.014 |
| NEIS-SS | 0.194 | 0.113 | −0.027–0.415 | 1.727 | 0.085 | 0.006 |
| Model 9 (DV = urge for material objects) | | | | | | |
| (intercept) | 1.989 | 0.945 | 0.132–3.846 | 2.104 | 0.036 | 0.009 |
| NEIS-S | −0.133 | 0.107 | −0.344–0.078 | −1.240 | 0.216 | 0.003 |
| NEIS-M | 0.535 | 0.105 | 0.329–0.741 | 5.100 | <0.001 | 0.053 |
| NEIS-SS | 0.222 | 0.117 | −0.008–0.451 | 1.898 | 0.058 | 0.007 |
| Model 10 (DV = urge for technology) | | | | | | |
| (intercept) | 1.493 | 1.040 | −0.550–3.536 | 1.436 | 0.152 | 0.004 |
| NEIS-S | −0.200 | 0.118 | −0.432–0.032 | −1.691 | 0.091 | 0.006 |
| NEIS-M | 0.510 | 0.115 | 0.284–0.737 | 4.422 | <0.001 | 0.039 |
| NEIS-SS | 0.245 | 0.129 | −0.008–0.498 | 1.906 | 0.057 | 0.007 |
| Model 11 (DV = under-stimulated) | | | | | | |
| (intercept) | 3.684 | 0.881 | 1.953–5.415 | 4.181 | <0.001 | 0.035 |
| NEIS-S | −0.007 | 0.100 | −0.204–0.190 | −0.071 | 0.943 | <0.001 |
| NEIS-M | 0.440 | 0.098 | 0.248–0.632 | 4.496 | <0.001 | 0.041 |
| NEIS-SS | 0.260 | 0.109 | 0.046–0.474 | 2.384 | 0.018† | 0.011 |
| Model 12 (DV = arousal) | | | | | | |
| (intercept) | 1.543 | 0.784 | 0.003–3.084 | 1.968 | 0.050 | 0.008 |
| NEIS-S | 0.097 | 0.089 | −0.078–0.272 | 1.092 | 0.275 | 0.002 |
| NEIS-M | 0.232 | 0.087 | 0.061–0.403 | 2.660 | 0.008† | 0.014 |
| NEIS-SS | −0.166 | 0.097 | −0.356–0.025 | −1.709 | 0.088 | 0.006 |
| behaviour | | | | | | |
| Model 13 (DV = cheating) | | | | | | |
| (intercept) | −0.310 | 1.292 | −2.849–2.228 | −0.240 | 0.810 | <0.001 |
| NEIS-S | 0.143 | 0.147 | −0.145–0.432 | 0.976 | 0.330 | 0.002 |
| NEIS-M | 0.464 | 0.143 | 0.182–0.746 | 3.237 | 0.001 | 0.021 |
| NEIS-SS | −0.132 | 0.160 | −0.446–0.182 | −0.826 | 0.409 | 0.001 |

not a significant predictor of people's experiences and behaviours during the lockdowns. By contrast, in Study 4, which focused on short-term input deprivation (i.e. spending time alone while entertaining oneself with one's thoughts), NEIS-M was the core predictor and was generally linked to more difficulty withstanding the deprivation (e.g. more boredom and intrusive thoughts, and more cheating). NEIS-SS was largely a non-significant predictor, and NEIS-S was even linked to positive experiences (i.e. higher meaningfulness and pleasantness). Therefore, Studies 3 and 4 hint that, depending on the duration of input deprivation, different needs for input may have varying links to psychological functioning.

## 9.1. Main contributions

The main overarching contribution of the present research is that it has integrated three lines of research that have so far led separate lives (i.e. materialism, social motives and needs, and sensation seeking) by

situating them under the common construct of external input. This general contribution has spawned several specific theoretical and methodological contributions.

First, the present research has uncovered that the needs for social, material and sensation seeking input are not mutually exclusive (e.g. having higher NEIS-M does not mean that NEIS-SS will be lower). Rather than counterbalancing each other, the three needs 'mirror' each other, given that they have a positive linear relationship. The current line of studies is the first to uncover this by systematically investigating the needs for external input in relation to an overarching construct. Previous research has rarely investigated the link between these needs, and even when correlations between them were computed, this was in many cases a 'side effect' of the main research question (e.g. when researchers studied whether they predict some dependent variable of interest, [29]). Indeed, no research implemented sophisticated psychometric tools (e.g. EFA and CFA, bifactor analysis) to understand the structure of external input regarding the three needs.

Second, this research has revealed that there is no one fundamental need for input. On a conceptual level, one could argue that NEIS-SS may be the most basic need, given that new or intense sensations can arise from interactions with both other people and material objects [19,86]. However, Studies 3 and 4 demonstrated that all NEIS needs were relevant to people's experiences of input deprivation, depending on its duration. Under long-term deprivation, NEIS-S and NEIS-SS were generally linked to more negative and less positive emotional experiences, whereas NEIS-M yielded no effects. By contrast, under short-term deprivation, the latter need was linked to negative experiences, whereas NEIS-SS was generally irrelevant, and NEIS-S was even associated with more positive experiences. The most plausible rationale behind this finding is that people who need others and novel or intense experiences generally do not need these stimuli every moment and it is enough if they assuage their needs on a regular basis [8,87]. By contrast, the need for material input may be manifested as a momentary urge to interact with objects such as smartphones [37]. In this respect, however, it is important to point out that the long-term input deprivation I investigated did not interfere with this urge as stringently as the short-term deprivation (i.e. despite COVID-19 restrictions, people could interact with material objects in their homes), which may have confounded the findings.

Third, on a methodological level, the main contribution of this research is the new scale—NEIS—that can be used to further investigate the need for external input and expand its theoretical and practical understanding. For example, it could be examined which biological and neural factors underpin NEIS-S, NEIS-M and NEIS-SS, and to what extent these needs predict important real-world outcomes (e.g. health and sustainability behaviours).

## 9.2. Main limitations

One limitation of the present research is that I focused specifically on material, social and sensation seeking input, but I neglected some other potentially relevant forms of input. In personality psychology, two such forms have been investigated: animals as external input [88] and nature as external input [89]. There is one main reason why I did not cover these constructs in the present research. I wanted to develop a general conceptualization of external input that all human beings are familiar with and that can be applied across all individuals. In that regard, there are many people who neither have pets nor live in the presence of animals, and there is a large diversity regarding exposure to nature (e.g. many people may live in cities with little nature). By contrast, most people live in the presence of some material objects, are exposed to various stimuli that can lead to new or intense sensations, and at least to some degree live surrounded by other human beings. Therefore, conceptualizing external input through these three dimensions makes this construct more universal and generalizable.

Another important limitation, as already suggested, is that Study 3 did not involve a strict long-term deprivation of all three forms of input, even if they were restricted compared with the normal. Moreover, some forms of input were hindered more than others (e.g. social and sensation seeking input may have been constrained more than material input). For ethical reasons, it is not possible to strictly deprive people of external input over a long period or restrict each form of input to a comparable degree, given the associated negative consequences. Moreover, by focusing on strict long-term input deprivation due to other circumstances (e.g. solitary confinement in prisons), it would be highly challenging to achieve a sample size needed for a well-powered study [90]. Given these considerations, it is likely that COVID-19 restrictions constitute one of the most optimal circumstances for investigating the link between the needs for external input and long-term input deprivation, despite the discussed limitations.

A related crucial limitation concerns several methodological weaknesses of Studies 3 and 4 which indicate that some of the main conclusions regarding these studies as a package (i.e. that the link between the needs for input and psychological experiences changes depending on input deprivation duration) must be taken with caution. Under ideal circumstances, Studies 3 and 4 would have been combined into a single study with a 2 (duration: long versus short term) × 2 (deprivation: absent versus present) between-subjects design. In other words, this study would consist of four conditions: one in which people would be deprived of input long-term, one in which they would be deprived of input short-term, one in which they would be exposed to input long-term and one in which they would be exposed to input short-term. Based on this study, one could then analyse both whether the link between the needs for social, material and sensation seeking input and various behaviours or emotional experiences differs as a function of deprivation duration and presence versus absence of input.

However, a closer look at this 'ideal' design reveals that it would face similar problems as the current design. First, given that the conditions in which external input is manipulated long-term involve a longitudinal design, participant attrition would make it difficult to compare these conditions with the ones in which the input is manipulated short-term. Indeed, it would be difficult to conclude whether any differences occurred due to the experimental manipulation or due to unequal participant samples caused by the attrition. Moreover, given that the long-term conditions would require substantially more time than the short-term conditions, it would be unclear whether any differences between them occurred due to the experimental manipulation or due to some unknown factor that took place over time.

When it comes to comparing the conditions in which absence versus presence of external input is manipulated, various methodological problems would also arise. Most importantly, it would be difficult to understand whether an effect is created by a presence versus absence of social, material or sensation seeking input. It appears that this issue could be resolved by independently manipulating either of these forms of input. However, the main obstacle in this regard is that the forms of input seem to be interlinked, and it is very difficult to manipulate only one of them while keeping others constant. In a pilot study I conducted that will be part of another paper I have not yet published, I presented participants with different scenarios about situations in which I manipulated each of the three input components, and then asked participants to rate the 'quantity' of each input component a situation contains. The result showed that in most cases increasing social input at least to some degree increases sensation seeking input and may also increase material input, and it is highly challenging to disentangle the three input components. Overall, based on these considerations, I conclude that even the 'ideal' design I describe would not represent an improvement compared with the current design, and a major methodological advancement would need to be undertaken to eliminate the weaknesses from which the present design of Studies 3 and 4 suffers.

The final limitation of the present research is that it uses correlational design. Whereas this type of design is usual in personality research and to some degree necessitated by the nature of personality scales that can mostly be analysed using correlational approaches, it is worthwhile to highlight the associated disadvantages. For example, it is not possible to determine whether the needs for external input cause changes in emotional experiences or merely predict them because they are correlated with the actual causal variables. Moreover, some of the shared variance among the subscales measuring the three needs for input could be due to method variance, which is a common issue in personality research (e.g. [91]).

## 9.3. Next steps

Considering that the present research offers preliminary insights into the needs for social, material and sensation seeking input as elements of the need for external input and into their link to experiences and behaviours during input deprivation, there are numerous next steps through which this line of research could be advanced. First, it will be necessary to develop a more rigorous methodology through which the quantity of each input component (i.e. social, material and sensation seeking) could be precisely manipulated, to understand how the three needs for input predict people's experiences and behaviours in situations with varying levels of external input. Methodological improvements will be necessary in this regard, given the weaknesses of the currently available methodological approaches that I discussed in the Main limitations section. Second, a real-world value of the needs for external input will need to be examined by investigating to what degree they shape behaviours such as consumption of material goods, use of natural resources and other activities that have implications for the most pressing real-world issues (e.g. climate change, inequality and health).

Ethics. The present research is aligned with the Research Ethics Policy and Code of Research Conduct of the Research Ethics Committee of the London School of Economics and Political Science and has received ethics approvals by this committee (REC ref. 1093 and 09586). Informed consent was obtained from all participants who took part in the present research.
Data accessibility. The study materials, data and analysis scripts used for this article can be accessed via the Open Science Framework (OSF): https://osf.io/mxezt/?view_only=4f65da59314443a89bd91156b2ae5ec2.

Electronic supplementary material is available online [92].
Author's contributions. D.K.: conceptualization, data curation, formal analysis, investigation, methodology, project administration, resources, writing—original draft and writing—review and editing.
Conflict of interest declaration. I declare I have no competing interests.
Funding. I received no funding for this study.

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
