## [Peer Review File · Royal Society Open Science]

Review History

RSOS-211373.R0 (Original submission)

Review form: Reviewer 1

Is the manuscript scientifically sound in its present form?

No

Are the interpretations and conclusions justified by the results?

No

Is the language acceptable?

No

Do you have any ethical concerns with this paper?

No

Have you any concerns about statistical analyses in this paper?

No

Recommendation?

Major revision is needed (please make suggestions in comments)

Comments to the Author(s)

I have just reviewed the manuscript: "To Sit Quietly in a Room Alone: The Psychology of Social, Material, and Sensation Seeking Input". This paper presents a new scale measuring sensation seeking vs. comfort with low input across three domains of life: social, material, and sensation seeking. Across four studies, the scale is developed, validated, and then tested in a number of well-powered samples collected online, mostly using MTurk and Prolific.

I had a number of reactions to the work. I'll list them here starting with the positives, and then expressing my concerns.

Positive reactions:

- 1) The Introduction and Discussion sections were clearly written and interesting.
- 2) Methodological decisions were thoughtfully made. On a number of occasions I had a concern that was then directly addressed in the manuscript. As an example, I appreciated the discussion on acquiescence bias and reverse coding, as well as the description of discriminant validity.
- 3) Study 2 convergent and discriminant measures were largely well-selected.
- 4) Study 4 would have been better as a individual difference X 2 conditions (with a manipulation of a high-stimulation / social lab time), but I still appreciated the approach although in essence, it was another correlational study.
- 5) Perhaps most important, the idea stands to make a useful contribution to the literature.

Concerns:

- 1) The studies are all essentially correlational. This is not a deal-breaker in my mind, but worth highlighting and recognizing that much of the shared variance among subscales could be due to method variance, including their simultaneous presentation to participants in one table (I assume).
- 2) The title sensationalizes the work. This research isn't about "To Sit Quietly in a Room Alone" - and actually without the manipulation I described above, the author cannot claim that the NEIS acts on this any differently than it would on 15 minutes spent in a group.
- 3) The paper oversells the work. I'd like to see a toning down of interpretations such as 'novel construct', since the work builds on existing constructs and informs them.
- 4) There was no mention of Coplan's work on loneliness, but it's relevant here. Coplan, R. J., Hipson, W. E., Archbell, K. A., Ooi, L. L., Baldwin, D., & Bowker, J. C. (2019). Seeking more solitude: Conceptualization, assessment, and implications of loneliness. *Personality and Individual Differences*, 148, 17-26.
- 5) Study 3 involve many assumptions, some of which felt like a stretch. I may have missed it, but was Study 2 only with living alone participants? For many, the period of lockdowns and COVID was extra-stimulating if they were home with family or children. Could the author discuss this and the implications for the research?
- 6) Study 4 as well, made a number of assumptions and results were attributed directly to short-term (Study 4) vs. long-term (Study 3) deprivation rather than a large number of other factors that could have explained each of the two studies. A Study 5 that replicates and integrates the previous two studies would have been very helpful.

Review form: Reviewer 2 (Thuy-vy Nguyen)

Is the manuscript scientifically sound in its present form?

Yes

Are the interpretations and conclusions justified by the results?

Yes

Is the language acceptable?

Yes

Do you have any ethical concerns with this paper?

No

Have you any concerns about statistical analyses in this paper?

No

Recommendation?

Accept with minor revision (please list in comments)

Comments to the Author(s)

There are many strengths to this manuscript:

1. First, the author provides very clear and thorough explanation of criteria the author used to identify items to include in their measure.
2. All data and codes are shared.
3. The author fully disclose that they did not have any initial hypotheses but generally describe some expectations that are sensible. I appreciate the transparency.

I have several points that I would like the author to address before I recommend acceptance of this manuscript.

First point is about the introduction:

The paper needs to situate itself better in relation to previous literatures on needs. I understand that the author is only interested in need for external stimuli. However, how are these needs being studied conceptually different from physical needs like needs for food, water, and sex, etc. Additionally, how are the needs being studied distinguished from basic psychological needs proposed by Deci & Ryan, or Sheldon and colleagues (see example of paper below). Since the literature already have physical/biological needs and psychological needs covered, I suggest one way to situate this work coherently with these literatures is to approach this from an experiential needs angle. This is just a suggestion, I am open to a different approach to address this comment; the main goal is to incorporate other literatures of need and make a case for why these 3 needs for external stimuli are different.

See: Sheldon, K. M., & Gunz, A. (2009). Psychological needs as basic motives, not just experiential requirements. *Journal of personality*, 77(5), 1467-1492.

Second point is about sensation seeking input:

Again, it might be worthwhile to clarify that the intention of the present research is not to focus on physical/biological needs but more experiential ones. I raised this point because originally I thought, by sensation seeking input, the author was referring to sensory inputs. Then, I found out that sensation seeking items capture novel experiences and stimuli rather than just stimulation of the senses. So I think this needs to be clarified.

Third point is about Study 4's design:

Study 4 yielded the weakest evidence of the pack; better evidence of those needs can be established through deliberately manipulating inputs corresponding with different subscales of NEIS.

I do not see Study 4 adds substantial insight into the evidence obtained from Studies 1-3, mainly because 15-minute free-time period is not meant to create deprivation experiences, particularly not social deprivation nor sensation deprivation. For social and sensation deprivation to be salient, I think it would take a longer period of time for participants to begin craving social interactions needing new stimulations in the room. One good example is from the work by Tomova et al (2020) that keeps people in the room for 10 hours, and different stimuli can be introduced or remove to distinguish between different needs.

Tomova, L., Wang, K. L., Thompson, T., Matthews, G. A., Takahashi, A., Tye, K. M., & Saxe, R. (2020). Acute social isolation evokes midbrain craving responses similar to hunger. *Nature Neuroscience*, 23(12), 1597-1605.

Therefore, I think Study 4 by itself does not add to the paper, unless more experiments to be added. For example, I would add an experiment with three conditions, each of which deprive people of each type of input. You can make the period longer: one condition without social input, one without novel/sensation seeking input, and one without material input (e.g., activities like 15-minute free-time period). This way, you will be able to demonstrate if someone who is high on one subscale on NEIS will correlate with outcomes in corresponding condition where that need is deprived. If it is not possible to add another experiment, then I think Study 4 should belong to a separate set of experiments instead of being part of this manuscript.

Thuy-vy Nguyen, PhD.

Decision letter (RSOS-211373.R0)

Dear Dr Krpan

The Editors assigned to your paper RSOS-211373 "To Sit Quietly in a Room Alone: The Psychology of Social, Material, and Sensation Seeking Input" have now received comments from reviewers and would like you to revise the paper in accordance with the reviewer comments and any comments from the Editors. Please note this decision does not guarantee eventual acceptance.

Please submit your revised manuscript and required files (see below) no later than 21 days from today's (ie 08-Nov-2021) date. Note: the ScholarOne system will 'lock' if submission of the revision is attempted 21 or more days after the deadline. If you do not think you will be able to meet this deadline please contact the editorial office immediately.

on behalf of Professor Geoff Haddock (Associate Editor) and Essi Viding (Subject Editor)
openscience@royalsociety.org

Associate Editor Comments to Author (Professor Geoff Haddock):

Associate Editor: 1

Comments to the Author:

Thank you for submitting your paper to RSOS. I have secured two reviews of your paper; both reviewers offer a thorough consideration of their impressions of the work. Overall, they see many strengths of the manuscript, but also raise a series of issues that they feel need to be addressed to further enhance the manuscript. My own reading converges with theirs – I like the ideas behind the paper, but there are some (relatively straightforward) concerns I would like you to address before making a final decision on the paper. As such, I'd like you to revise and resubmit the paper.

The reviewers' comments are clear, and for the sake of brevity I will not repeat them here.

Needless to say, I anticipate that their concerns would be addressed in a revision. I fully agree with both reviewers that the paper would benefit from being better situated in the context of existing work - both reviewers offer some helpful suggestions. I also share one reviewer's concern about re-thinking the title, and slightly toning down some of the interpretations drawn from the studies. Aside from the reviewers' points, I also wondered about the number of exclusions for Studies 2 and 4 - please add that information to the supplemental information.

Finally, it seems to me that the general discussion would benefit from a more detailed consideration of what you perceive as the most important next steps, based upon the findings that you present in the paper.

As noted above, I expect that these revisions should be relatively straightforward to complete.

When resubmitting the paper, please provide a cover letter outlining the revisions that have been made.

Thank you for submitting the paper to RSOS, and I wish you continued success in all of your research activities.

Sincerely,
Professor Geoff Haddock

Reviewer comments to Author:

Reviewer: 1

Comments to the Author(s)

I have just reviewed the manuscript: "To Sit Quietly in a Room Alone: The Psychology of Social, Material, and Sensation Seeking Input". This paper presents a new scale measuring sensation seeking vs. comfort with low input across three domains of life: social, material, and sensation seeking. Across four studies, the scale is developed, validated, and then tested in a number of well-powered samples collected online, mostly using MTurk and Prolific.

I had a number of reactions to the work. I'll list them here starting with the positives, and then expressing my concerns.

Positive reactions:

- 1) The Introduction and Discussion sections were clearly written and interesting.
- 2) Methodological decisions were thoughtfully made. On a number of occasions I had a concern that was then directly addressed in the manuscript. As an example, I appreciated the discussion on acquiescence bias and reverse coding, as well as the description of discriminant validity.
- 3) Study 2 convergent and discriminant measures were largely well-selected.
- 4) Study 4 would have been better as a individual difference X 2 conditions (with a manipulation of a high-stimulation / social lab time), but I still appreciated the approach although in essence, it was another correlational study.
- 5) Perhaps most important, the idea stands to make a useful contribution to the literature.

Concerns:

- 1) The studies are all essentially correlational. This is not a deal-breaker in my mind, but worth highlighting and recognizing that much of the shared variance among subscales could be due to method variance, including their simultaneous presentation to participants in one table (I assume).
- 2) The title sensationalizes the work. This research isn't about "To Sit Quietly in a Room Alone" - and actually without the manipulation I described above, the author cannot claim that the NEIS acts on this any differently than it would on 15 minutes spent in a group.
- 3) The paper oversells the work. I'd like to see a toning down of interpretations such as 'novel construct', since the work builds on existing constructs and informs them.
- 4) There was no mention of Coplan's work on loneliness, but it's relevant here. Coplan, R. J., Hipson, W. E., Archbell, K. A., Ooi, L. L., Baldwin, D., & Bowker, J. C. (2019). Seeking more solitude: Conceptualization, assessment, and implications of loneliness. *Personality and Individual Differences*, 148, 17-26.
- 5) Study 3 involve many assumptions, some of which felt like a stretch. I may have missed it, but was Study 2 only with living alone participants? For many, the period of lockdowns and COVID was extra-stimulating if they were home with family or children. Could the author discuss this and the implications for the research?
- 6) Study 4 as well, made a number of assumptions and results were attributed directly to short-term (Study 4) vs. long-term (Study 3) deprivation rather than a large number of other factors that could have explained each of the two studies. A Study 5 that replicates and integrates the previous two studies would have been very helpful.

Reviewer: 2

Comments to the Author(s)

There are many strengths to this manuscript:

1. First, the author provides very clear and thorough explanation of criteria the author used to identify items to include in their measure.
2. All data and codes are shared.
3. The author fully disclose that they did not have any initial hypotheses but generally describe some expectations that are sensible. I appreciate the transparency.

I have several points that I would like the author to address before I recommend acceptance of this manuscript.

First point is about the introduction:

The paper needs to situate itself better in relation to previous literatures on needs. I understand that the author is only interested in need for external stimuli. However, how are these needs being studied conceptually different from physical needs like needs for food, water, and sex, etc. Additionally, how are the needs being studied distinguished from basic psychological needs proposed by Deci & Ryan, or Sheldon and colleagues (see example of paper below). Since the literature already have physical/biological needs and psychological needs covered, I suggest one way to situate this work coherently with these literatures is to approach this from an experiential needs angle. This is just a suggestion, I am open to a different approach to address this comment; the main goal is to incorporate other literatures of need and make a case for why these 3 needs for external stimuli are different.

See: Sheldon, K. M., & Gunz, A. (2009). Psychological needs as basic motives, not just experiential requirements. *Journal of personality*, 77(5), 1467-1492.

Second point is about sensation seeking input:

Again, it might be worthwhile to clarify that the intention of the present research is not to focus on physical/biological needs but more experiential ones. I raised this point because originally I thought, by sensation seeking input, the author was referring to sensory inputs. Then, I found out that sensation seeking items capture novel experiences and stimuli rather than just stimulation of the senses. So I think this needs to be clarified.

Third point is about Study 4's design:

Study 4 yielded the weakest evidence of the pack; better evidence of those needs can be established through deliberately manipulating inputs corresponding with different subscales of NEIS.

I do not see Study 4 adds substantial insight into the evidence obtained from Studies 1-3, mainly because 15-minute free-time period is not meant to create deprivation experiences, particularly not social deprivation nor sensation deprivation. For social and sensation deprivation to be salient, I think it would take a longer period of time for participants to begin craving social interactions needing new stimulations in the room. One good example is from the work by Tomova et al (2020) that keeps people in the room for 10 hours, and different stimuli can be introduced or remove to distinguish between different needs.

Tomova, L., Wang, K. L., Thompson, T., Matthews, G. A., Takahashi, A., Tye, K. M., & Saxe, R. (2020). Acute social isolation evokes midbrain craving responses similar to hunger. *Nature Neuroscience*, 23(12), 1597-1605.

Therefore, I think Study 4 by itself does not add to the paper, unless more experiments to be added. For example, I would add an experiment with three conditions, each of which deprive

people of each type of input. You can make the period longer: one condition without social input, one without novel/sensation seeking input, and one without material input (e.g., activities like 15-minute free-time period). This way, you will be able to demonstrate if someone who is high on one subscale on NEIS will correlate with outcomes in corresponding condition where that need is deprived. If it is not possible to add another experiment, then I think Study 4 should belong to a separate set of experiments instead of being part of this manuscript.

Thuy-vy Nguyen, PhD.

===PREPARING YOUR MANUSCRIPT===

'clean' version of the new manuscript that incorporates the changes made, but does not highlight them. This version will be used for typesetting if your manuscript is accepted.

If you have been asked to revise the written English in your submission as a condition of publication, you must do so, and you are expected to provide evidence that you have received language editing support. The journal would prefer that you use a professional language editing service and provide a certificate of editing, but a signed letter from a colleague who is a fluent speaker of English is acceptable. Note the journal has arranged a number of discounts for authors using professional language editing services (<https://royalsociety.org/journals/authors/benefits/language-editing/>).

===PREPARING YOUR REVISION IN SCHOLARONE===

Please ensure that you include a summary of your paper at Step 2 'Type, Title, & Abstract'. This should be no more than 100 words to explain to a non-scientific audience the key findings of your

research. This will be included in a weekly highlights email circulated by the Royal Society press office to national UK, international, and scientific news outlets to promote your work.

Author's Response to Decision Letter for (RSOS-211373.R0)

See Appendix A.

Decision letter (RSOS-211373.R1)

Dear Dr Krpan,

It is a pleasure to accept your manuscript entitled "Exploring the Need for External Input Through the Prism of Social, Material, and Sensation Seeking Input" in its current form for publication in Royal Society Open Science. The comments from the Editors are included at the foot of this letter.

Please send to the editorial office an editable version of your accepted manuscript, and individual files for each figure and table included in your manuscript. You can send these in a zip folder if more convenient. Failure to provide these files may delay the processing of your proof.

on behalf of Professor Geoff Haddock (Associate Editor) and Essi Viding (Subject Editor)
openscience@royalsociety.org

Associate Editor Comments to Author (Professor Geoff Haddock):

Thank you for submitting the revised version of your paper; I very much appreciate the attention devoted to the revisions. Overall, the revisions do a very good job of dealing with the points raised by myself and the reviewer. As a result, I am pleased to accept your paper for publication in RSOS. Compliments on this interesting and important piece of work.

Sincerely,
Geoff Haddock
Associate Editor, RSOS

Appendix A

ASSOCIATE EDITOR'S COMMENTS

COMMENT 1: Thank you for submitting your paper to RSOS. I have secured two reviews of your paper; both reviewers offer a thorough consideration of their impressions of the work. Overall, they see many strengths of the manuscript, but also raise a series of issues that they feel need to be addressed to further enhance the manuscript. My own reading converges with theirs – I like the ideas behind the paper, but there are some (relatively straightforward) concerns I would like you to address before making a final decision on the paper. As such, I'd like you to revise and resubmit the paper.

RESPONSE: Dear Professor Haddock, I would like to thank you and the reviewers for reading my manuscript thoroughly and giving me extremely useful comments that were helpful in revising it. To do justice to the efforts you have put in to help me improve the paper, I have considered each comment thoughtfully. I hope the revisions described below reflect this approach and address your concerns.

COMMENT 2: The reviewers' comments are clear, and for the sake of brevity I will not repeat them here. Needless to say, I anticipate that their concerns would be addressed in a revision. I fully agree with both reviewers that the paper would benefit from being better situated in the context of existing work - both reviewers offer some helpful suggestions.

RESPONSE: Thank you for raising this issue. I agree that the paper could have been better situated in the context of existing work. To address this concern, I adopted the recommendations suggested by both reviewers.

Namely, in line with Reviewer 1's suggestion, I included a reference to aloneliness in the overview of constructs that tap into social input as follows (see p.4 in the revised manuscript and my response to Reviewer 1's Comment 5): "Various individual difference constructs that at least to some degree tap into the three needs for input have been developed. Concerning social input, these constructs involve the need to belong (Baumeister & Leary, 1995); fundamental social motives and affiliation motivation (Hill, 1987; Neel, Kenrick, White, & Neuberg, 2016); preference for solitude (Burger, 1995); attachment (Brennan & Shaver, 1995); dependency (Pincus & Gurtman, 1995); collectivism-individualism (Triandis & Gelfand, 1998); communal orientation (Clark, Oullette, Powell, & Milberg, 1987); loneliness (Russell, Peplau, & Cutrona, 1980); **aloneliness** (Coplan et al., 2019), and relatedness (Johnston & Finney, 2010)."

Moreover, in line with Reviewer 2's suggestion (Comments 2 & 3), I have inserted the following paragraph in the section "The Needs for Social, Material, and Sensation Seeking Input" (p.5) to sufficiently situate the paper in relation to previous literature on needs:

"Considering that I approach the construct of external input through the prism of needs, it is necessary to situate it in relation to previous literature on needs. I do not conceptualize the needs for social, material, and sensation seeking input as biological or physical needs that

typically refer to the basic requirements for maintaining human life, such as food or shelter (e.g., Maslow, 1943; Tay & Diener, 2011). Indeed, even if the expressions that I use in relation to external input, such as “any physical stimulation that can be detected by the senses”, may be evocative of physical needs, I utilize these expressions because I approach external input from a materialist perspective rather than because I write about physical needs. Given that I define the needs for social, material, and sensation seeking input in terms of how much people require the corresponding input components to maintain optimal psychological functioning, these needs can be classified as experiential needs (i.e., needs that are important for wellbeing; Sheldon & Gunz, 2009). However, based on previous research, it would be difficult to argue whether these three needs are basic psychological needs in the sense that they all need to be met for a person to experience high wellbeing, as would be the case for the needs such as autonomy, competence, and relatedness stemming from the self-determination theory (Deci & Ryan, 2008). Whereas investigating this would be an interesting topic for future research, it is not the focus of the present article.”

COMMENT 3: I also share one reviewer’s concern about re-thinking the title, and slightly toning down some of the interpretations drawn from the studies.

RESPONSE: I agree with you and Reviewer 1 that the title sensationalizes the work, given the implications of Study 4. For that reason, I have changed the title from “To Sit Quietly in a Room Alone: The Psychology of Social, Material, and Sensation Seeking Input” to “Exploring the Need for External Input Through the Prism of Social, Material, and Sensation Seeking Input”. The new title more accurately reflects the core content of the paper, which revolves around examining the need for external input through the prism of social, material, and sensation seeking input.

Concerning toning down some of the interpretations drawn from the studies, I have made several changes, including toning down some of the language and more critically discussing Studies 3 and 4. These comprehensive changes can be seen in my response to Comments 4 and 7 by Reviewer 2.

COMMENT 4: Aside from the reviewers’ points, I also wondered about the number of exclusions for Studies 2 and 4 – please add that information to the supplemental information.

RESPONSE: Thank you for raising this issue. In the previous manuscript, it was indeed not clear how many participants were excluded from statistical analyses in each study. I have now updated this information in the section “Participants and Exclusion Criteria” in Supplementary Materials (pp.4-5). For Study 1, I wrote “In Sample 1, 85 participants were therefore excluded from analyses; in Sample 2, 125 participants; and in Sample 3, 47 participants” (p.4). For Study 2, I wrote “Therefore, 49 participants were excluded from analyses” (p.5). For Study 3, I wrote “Therefore, out of the 2756 participants who completed

the entire study, 764 were excluded from analyses” (p.5). Finally, for Study 4 I wrote “Overall, 51 participants were excluded from analyses.” (p.5). Moreover, to make sure it is clear that Table S1 in Supplementary Materials (p.4) contains the summary for all participants recruited in each study, I expanded its title to “Sample Size and Demographics for all Participants (i.e., including those excluded from and included in statistical analyses) Recruited in Studies 1-4.”

COMMENT 5: Finally, it seems to me that the general discussion would benefit from a more detailed consideration of what you perceive as the most important next steps, based upon the findings that you present in the paper.

RESPONSE: I fully agree that the general discussion would benefit from a more detailed consideration of what I perceive as the most important next steps. To address your comment, I have therefore included the section “Next Steps” in the general discussion as follows (pp.44-45):

“Considering that the present research offers preliminary insights into the needs for social, material, and sensation seeking input as elements of the need for external input and into their link to experiences and behaviours during input deprivation, there are numerous next steps through which this line of research could be advanced. First, it will be necessary to develop a more rigorous methodology through which the quantity of each input component (i.e., social, material, and sensation seeking) could be precisely manipulated, to understand how the three needs for input predict people’s experiences and behaviours in situations with varying levels of external input. Methodological improvements will be necessary in this regard, given the weaknesses of the currently available methodological approaches that I discussed in the Main Limitations section. Second, a real-world value of the needs for external input will need to be examined by investigating to what degree they shape behaviours such as consumption of material goods, use of natural resources, and other activities that have implications for the most pressing real-world issues (e.g., climate change, inequality, health).”

COMMENT 6: As noted above, I expect that these revisions should be relatively straightforward to complete. When resubmitting the paper, please provide a cover letter outlining the revisions that have been made. Thank you for submitting the paper to RSOS, and I wish you continued success in all of your research activities.

RESPONSE: Thank you once again for the extensive feedback, I appreciate that you and the reviewers took the time to help me improve the manuscript.

REVIEWER 1'S COMMENTS

COMMENT 1: I have just reviewed the manuscript: "To Sit Quietly in a Room Alone: The Psychology of Social, Material, and Sensation Seeking Input". This paper presents a new scale measuring sensation seeking vs. comfort with low input across three domains of life: social, material, and sensation seeking. Across four studies, the scale is developed, validated, and then tested in a number of well-powered samples collected online, mostly using MTurk and Prolific.

I had a number of reactions to the work. I'll list them here starting with the positives, and then expressing my concerns.

Positive reactions:

- 1) The Introduction and Discussion sections were clearly written and interesting.
- 2) Methodological decisions were thoughtfully made. On a number of occasions I had a concern that was then directly addressed in the manuscript. As an example, I appreciated the discussion on acquiescence bias and reverse coding, as well as the description of discriminant validity.
- 3) Study 2 convergent and discriminant measures were largely well-selected.
- 4) Study 4 would have been better as a individual difference X 2 conditions (with a manipulation of a high-stimulation / social lab time), but I still appreciated the approach although in essence, it was another correlational study.
- 5) Perhaps most important, the idea stands to make a useful contribution to the literature.

RESPONSE: First of all, thank you for thoroughly reading the manuscript and appreciating its contributions. I am grateful that you have read the manuscript thoroughly and gave me highly useful feedback to improve it. In subsequent comments, I outline in detail how I addressed your concerns.

COMMENT 2: Concerns: 1) The studies are all essentially correlational. This is not a deal-breaker in my mind, but worth highlighting and recognizing that much of the shared variance among subscales could be due to method variance, including their simultaneous presentation to participants in one table (I assume).

RESPONSE: Thank you for raising this issue. To address your comment, I have included the following paragraph in the Main Limitations section (**p.44**):

"The final limitation of the present research is that it uses correlational design. Whereas this type of design is usual in personality research and to some degree necessitated by the nature of personality scales that can mostly be analysed using correlational approaches, it is worthwhile to highlight the associated disadvantages. For example, it is not possible to determine whether the needs for external input cause changes in emotional experiences or merely predict them because they are correlated with the actual causal variables. Moreover, some of the shared variance among the subscales measuring the three needs for input could

be due to method variance, which is a common issue in personality research (e.g., Biderman, Nguyen, Cunningham, & Ghorbani, 2011).”

COMMENT 3: 2) The title sensationalizes the work. This research isn’t about “To Sit Quietly in a Room Alone” – and actually without the manipulation I described above, the author cannot claim that the NEIS acts on this any differently than it would on 15 minutes spent in a group.

RESPONSE: I agree that the title sensationalizes the work, given the limitations of Study 4 that you raise. For that reason, I have changed the title from “To Sit Quietly in a Room Alone: The Psychology of Social, Material, and Sensation Seeking Input” to “Exploring the Need for External Input Through the Prism of Social, Material, and Sensation Seeking Input”. The new title more accurately reflects the core content of the paper, which revolves around examining the need for external input through the prism of social, material, and sensation seeking input.

COMMENT 4: 3) The paper oversells the work. I’d like to see a toning down of interpretations such as ‘novel construct’, since the work builds on existing constructs and informs them.

RESPONSE: I agree with you that in some aspects of the paper I oversold the work. To address this issue, in addition to changing the title (as indicated in my response to Comment 3), I changed most expressions referring to a “novel construct”. Most importantly, I changed the previous conclusion at the end of the abstract from “Overall, this research established a novel construct that has fundamental implications for experiences and actions in a range of different contexts” to a more cautious conclusion (**see p.2**): “Overall, this research indicates that the needs for social, material, and sensation seeking input may have fundamental implications for experiences and actions in a range of different contexts.”

I left the word “novel” only in the following sentence (**p.2**): “The present research established a novel conceptualization of this construct by investigating it in relation to the needs for material, social, and sensation seeking input, and by testing whether these needs predict psychological functioning during long- and short-term input deprivation.” I think using the word “novel” here is appropriate because I am not claiming that I established a novel construct, but that I established a novel conceptualization of external input, which is accurate because I investigated the needs for social, material, and sensation seeking input through the prism of a common construct, which has not been done before. However, I am open to removing the word “novel” from this sentence as well if you think that this oversells the work.

COMMENT 5: 4) There was no mention of Coplan’s work on aloneliness, but it’s relevant here.

Coplan, R. J., Hipson, W. E., Archbell, K. A., Ooi, L. L., Baldwin, D., & Bowker, J. C. (2019). Seeking more solitude: Conceptualization, assessment, and implications of aloneliness. *Personality and Individual Differences*, 148, 17-26.

RESPONSE: Thank you for pointing this out. In the previous version of the manuscript, I have failed to refer to this work because it was published after I have already conducted the initial studies in which the need for external input scale (NEIS) was validated, and hence I did not consider this scale when devising the NEIS items. However, I completely agree that it is important in the context of social input, and hence I included it in the overview of constructs that tap into social input as follows (see p.4): “Various individual difference constructs that at least to some degree tap into the three needs for input have been developed. Concerning social input, these constructs involve the need to belong (Baumeister & Leary, 1995); fundamental social motives and affiliation motivation (Hill, 1987; Neel, Kenrick, White, & Neuberg, 2016); preference for solitude (Burger, 1995); attachment (Brennan & Shaver, 1995); dependency (Pincus & Gurtman, 1995); collectivism-individualism (Triandis & Gelfand, 1998); communal orientation (Clark, Oullette, Powell, & Milberg, 1987); loneliness (Russell, Peplau, & Cutrona, 1980); **aloneliness** (Coplan et al., 2019), and relatedness (Johnston & Finney, 2010).”

COMMENT 6: 5) Study 3 involve many assumptions, some of which felt like a stretch. I may have missed it, but was Study 3 only with living alone participants? For many, the period of lockdowns and COVID was extra-stimulating if they were home with family or children. Could the author discuss this and the implications for the research?

RESPONSE: Thank you for raising this issue. In this case, it is my fault that I did not clearly state that Study 3 did not focus only on living alone participants, but on a wide range of participants regardless of their household size. Indeed, variables such as household size were measured as covariates and controlled for in statistical analyses to ensure that the results are not confounded by variables that may affect people’s experiences of the lockdowns. To make sure there is no unclarity that could create misunderstandings in this regard, in the revised manuscript I expanded a paragraph describing participants and procedure in Study 3 as follows:

“In part 1, participants first completed the consent form, after which demographics (nationality and country of residence) and covariates were assessed. In that context, it is important to emphasize that the study did not focus on a particular type of participants (e.g., those who lived alone or were used to spending time at home). Instead, various variables relevant to psychological experiences and compliance regarding COVID-19 were measured as covariates and controlled for in statistical analyses to avoid potential confounding effects (see the Results section below). Examples of covariates include people’s distancing history (i.e., when they first started practicing social distancing), household size (i.e., how many people lived together with the participant in the same household), living situation (i.e.,

whether participants' living situation allowed them to comply with social distancing), or being used to spending time at home (i.e., how many full days participants would typically spend at home before the COVID-19 pandemic started). A comprehensive list and description of all the covariates measured is available in SM (pp.42-43).”

COMMENT 7: 6) Study 4 as well, made a number of assumptions and results were attributed directly to short-term (Study 4) vs. long-term (Study 3) deprivation rather than a large number of other factors that could have explained each of the two studies. A Study 5 that replicates and integrates the previous two studies would have been very helpful.

RESPONSE: I agree that Studies 3 and 4 as a package made a number of assumptions and have several limitations that hamper interpretability of the findings. I have decided to deal with this issue by adding a comprehensive discussion of these limitations in the Main Limitations section (pp.42-44) as follows:

“A related crucial limitation concerns several methodological weaknesses of Studies 3-4 which indicate that some of the main conclusions regarding these studies as a package (i.e., that the link between the needs for input and psychological experiences changes depending on input deprivation duration) must be taken with caution. Under ideal circumstances, Studies 3 and 4 would have been combined into a single study with a 2 (duration: long vs. short term) x 2 (deprivation: absent vs. present) between-subjects design. In other words, this study would consist of 4 conditions: one in which people would be deprived of input long-term, one in which they would be deprived of input short-term, one in which they would be exposed to input long-term, and one in which they would be exposed to input short-term. Based on this study, one could then analyse both whether the link between the needs for social, material, and sensation seeking input and various behaviours or emotional experiences differs as a function of deprivation duration and presence vs. absence of input.

However, a closer look at this “ideal” design reveals that it would face similar problems as the current design. First, given that the conditions in which external input is manipulated long-term involve a longitudinal design, participant attrition would make it difficult to compare these conditions with the ones in which the input is manipulated short-term. Indeed, it would be difficult to conclude whether any differences occurred due to the experimental manipulation or due to unequal participant samples caused by the attrition. Moreover, given that the long-term conditions would require substantially more time than the short-term conditions, it would be unclear whether any differences between them occurred due to the experimental manipulation or due to some unknown factor that took place over time.

When it comes to comparing the conditions in which absence versus presence of external input is manipulated, various methodological problems would also arise. Most importantly, it would be difficult to understand whether an effect is created by a presence versus absence of social, material, or sensation seeking input. It appears that this issue could be resolved by independently manipulating either of these forms of input. However, the main obstacle in this regard is that the forms of input seem to be interlinked, and it is very difficult to

manipulate only one of them while keeping others constant. In a pilot study I conducted that will be part of another paper I have not yet published, I presented participants with different scenarios about situations in which I manipulated each of the three input components, and then asked participants to rate the “quantity” of each input component a situation contains. The result showed that in most cases increasing social input at least to some degree increases sensation seeking input and may also increase material input, and it is highly challenging to disentangle the three input components. Overall, based on these considerations, I conclude that even the “ideal” design I describe would not represent an improvement compared to the current design, and a major methodological advancement would need to be undertaken to eliminate the weaknesses from which the present design of Studies 3-4 suffers.”

Overall, even if the methodology I used in Studies 3-4 has several limitations, this discussion of the limitations shows that I have considered several other designs and did not find a methodology that would solve the problems. Therefore, devising such a methodology is my next aim in developing this line of research.

In addition to the comprehensive discussion of the limitations, to tone down assumptions regarding Studies 3-4, I have changed several expressions describing these studies to make sure the language used is not causal but is reflective of the correlational nature of the studies. Therefore, the following changes were made:

The sentence “The present research established a novel conceptualization of this construct by investigating it in relation to the needs for material, social, and sensation seeking input, and by testing the consequences of these needs for psychological functioning during long- and short-term input deprivation.” was changed into “The present research established a novel conceptualization of this construct by investigating it in relation to the needs for material, social, and sensation seeking input, and by testing whether these needs predict psychological functioning during long- and short-term input deprivation.” (p.2)

The sentence “It was established that the three needs constitute different dimensions of an overarching construct (i.e., need for external input), that the needs for social and sensation seeking input have negative consequences for people’s experiences of long-term input deprivation (i.e., COVID-19 restrictions), and that the need for material input negatively predicts the experiences of short-term input deprivation (i.e., sitting in a chair without doing anything else but thinking).” was changed into “It was established that the three needs constitute different dimensions of an overarching construct (i.e., need for external input). The research also suggested that the needs for social and sensation seeking input are negatively linked to people’s experiences of long-term input deprivation (i.e., COVID-19 restrictions), and that the need for material input may negatively predict the experiences of short-term input deprivation (i.e., sitting in a chair without doing anything else but thinking).” (p.2).

The sentence “Overall, this research established a novel construct that has fundamental implications for experiences and actions in a range of different contexts.” was changed into “Overall, this research indicates that the needs for social, material, and sensation seeking

input may have fundamental implications for experiences and actions in a range of different contexts.” (p.2)

The sentence “Moreover, even if NEIS-SS had various negative consequences in Study 3, it was largely a non-significant predictor of experiences and behaviour in the present study.” was changed into “Moreover, even if NEIS-SS was negatively linked to wellbeing in Study 3, it was largely a non-significant predictor of experiences and behaviour in the present study.” (p.38)

The sentence “The present research offered preliminary insights into the psychology of external input by examining this construct in relation to the needs for material, social, and sensation seeking input (RQ1), and by probing the consequences of these needs for psychological functioning during long- and short-term input deprivation (RQ2).” was changed into “The present research offered preliminary insights into the psychology of external input by examining this construct in relation to the needs for material, social, and sensation seeking input (RQ1), and by probing the link between these needs and psychological functioning during long- and short-term input deprivation (RQ2).” (p.39)

The sentence “Regarding RQ2, Study 3 showed that NEIS-S and NEIS-SS had negative consequences for people’s emotional experiences during COVID-19 lockdowns.” was changed into “Regarding RQ2, Study 3 showed that NEIS-S and NEIS-SS had negative implications for people’s wellbeing during COVID-19 lockdowns, as they were negative predictors of positive emotional experiences and positive predictors of negative emotional experiences.” (p.39)

The sentences “NEIS-SS was largely a non-significant predictor, and NEIS-S even had positive consequences (e.g., it was linked to higher meaningfulness and pleasantness). Therefore, studies 3-4 established that, depending on the duration of input deprivation, different needs for input have different consequences for psychological functioning.” were changed into “NEIS-SS was largely a non-significant predictor, and NEIS-S was even linked to positive experiences (i.e., higher meaningfulness and pleasantness). Therefore, studies 3-4 hint that, depending on the duration of input deprivation, different needs for input may have varying links to psychological functioning.” (pp.39-40)

The sentences “Under long-term deprivation, NEIS-S and NEIS-SS generally had negative consequences, whereas NEIS-M yielded no effects. In contrast, under short-term deprivation, the latter need had the most negative consequences, whereas NEIS-SS was generally irrelevant and NEIS-S even positively predicted people’s experiences.” were changed into “Under long-term deprivation, NEIS-S and NEIS-SS were generally linked to more negative and less positive emotional experiences, whereas NEIS-M yielded no effects. In contrast, under short-term deprivation, the latter need was linked to negative experiences, whereas NEIS-SS was generally irrelevant, and NEIS-S was even associated with more positive experiences.” (pp.40-41)

REVIEWER 2'S COMMENTS

COMMENT 1: There are many strengths to this manuscript:

1. First, the author provides very clear and thorough explanation of criteria the author used to identify items to include in their measure.
2. All data and codes are shared.
3. The author fully disclose that they did not have any initial hypotheses but generally describe some expectations that are sensible. I appreciate the transparency.

RESPONSE: For the start, I would like to thank you for thoroughly reading the manuscript and recognizing its strengths as well as pointing out its weaknesses. Your comments challenged me to reconsider some of my initial assumptions and conclusions, and I am grateful for your input that helped me to improve the quality of the manuscript. In the subsequent comments, I address the issues you raised.

COMMENT 2: I have several points that I would like the author to address before I recommend acceptance of this manuscript.

First point is about the introduction:

The paper needs to situate itself better in relation to previous literatures on needs. I understand that the author is only interested in need for external stimuli. However, how are these needs being studied conceptually different from physical needs like needs for food, water, and sex, etc. Additionally, how are the needs being studied distinguished from basic psychological needs proposed by Deci & Ryan, or Sheldon and colleagues (see example of paper below). Since the literature already have physical/biological needs and psychological needs covered, I suggest one way to situate this work coherently with these literatures is to approach this from an experiential needs angle. This is just a suggestion, I am open to a different approach to address this comment; the main goal is to incorporate other literatures of need and make a case for why these 3 needs for external stimuli are different.

See: Sheldon, K. M., & Gunz, A. (2009). Psychological needs as basic motives, not just experiential requirements. *Journal of personality*, 77(5), 1467-1492.

RESPONSE: I think this is an extremely valuable comment. I agree that, in the initial version of the manuscript, I did not sufficiently situate the needs for social, material, and sensation seeking input in relation to previous literature on needs. I would like to thank you for the pointers you gave me in this regard. To address this issue, I have inserted the following paragraph in the section "The Needs for Social, Material, and Sensation Seeking Input" (p.5):

"Considering that I approach the construct of external input through the prism of needs, it is necessary to situate it in relation to previous literature on needs. I do not conceptualize the needs for social, material, and sensation seeking input as biological or physical needs that typically refer to the basic requirements for maintaining human life, such as food or shelter (e.g., Maslow, 1943; Tay & Diener, 2011). Indeed, even if the expressions that I use in

relation to external input, such as “any physical stimulation that can be detected by the senses”, may be evocative of physical needs, I utilize these expressions because I approach external input from a materialist perspective rather than because I write about physical needs. Given that I define the needs for social, material, and sensation seeking input in terms of how much people require the corresponding input components to maintain optimal psychological functioning, these needs can be classified as experiential needs (i.e., needs that are important for wellbeing; Sheldon & Gunz, 2009). However, based on previous research, it would be difficult to argue whether these three needs are basic psychological needs in the sense that they all need to be met for a person to experience high wellbeing, as would be the case for the needs such as autonomy, competence, and relatedness stemming from the self-determination theory (Deci & Ryan, 2008). Whereas investigating this would be an interesting topic for future research, it is not the focus of the present article.”

COMMENT 3: Second point is about sensation seeking input:

Again, it might be worthwhile to clarify that the intention of the present research is not to focus on physical/biological needs but more experiential ones. I raised this point because originally I thought, by sensation seeking input, the author was referring to sensory inputs. Then, I found out that sensation seeking items capture novel experiences and stimuli rather than just stimulation of the senses. So I think this needs to be clarified.

RESPONSE: Thank you for raising this issue. In the revised manuscript, I have addressed your comment in the new paragraph (p.5) that I created in response to your previous comment. In particular, the following section addresses the present comment (p.5): “I do not conceptualize the needs for social, material, and sensation seeking input as biological or physical needs that typically refer to the basic requirements for maintaining human life, such as food or shelter (e.g., Maslow, 1943; Tay & Diener, 2011). Indeed, even if the expressions that I use in relation to external input, such as “any physical stimulation that can be detected by the senses”, may be evocative of physical needs, I utilize these expressions because I approach external input from a materialist perspective rather than because I write about physical needs.”

COMMENT 4:

Third point is about Study 4’s design:

Study 4 yielded the weakest evidence of the pack; better evidence of those needs can be established through deliberately manipulating inputs corresponding with different subscales of NEIS.

I do not see Study 4 adds substantial insight into the evidence obtained from Studies 1-3, mainly because 15-minute free-time period is not meant to create deprivation experiences, particularly not social deprivation nor sensation deprivation. For social and sensation deprivation to be salient, I think it would take a longer period of time for participants to begin craving social interactions needing new stimulations in the room. One good example is from the work by Tomova et al (2020) that keeps people in the room for 10 hours, and different

stimuli can be introduced or removed to distinguish between different needs.

Tomova, L., Wang, K. L., Thompson, T., Matthews, G. A., Takahashi, A., Tye, K. M., & Saxe, R. (2020). Acute social isolation evokes midbrain craving responses similar to hunger. *Nature Neuroscience*, 23(12), 1597-1605.

Therefore, I think Study 4 by itself does not add to the paper, unless more experiments are added. For example, I would add an experiment with three conditions, each of which deprives people of each type of input. You can make the period longer: one condition without social input, one without novel/sensation seeking input, and one without material input (e.g., activities like 15-minute free-time period). This way, you will be able to demonstrate if someone who is high on one subscale on NEIS will correlate with outcomes in corresponding conditions where that need is deprived. If it is not possible to add another experiment, then I think Study 4 should belong to a separate set of experiments instead of being part of this manuscript.

RESPONSE: I fully agree with your comments that Study 4 has several weaknesses that to some degree limit its implications. However, before conducting the study, I considered other possible designs, and I came to the conclusion that they would not be a significant improvement compared to the current Study 4. To address your comment, in the revised manuscript I have therefore extensively discussed these considerations in the section “Main Limitations” as follows (pp.42-44):

“A related crucial limitation concerns several methodological weaknesses of Studies 3-4 which indicate that some of the main conclusions regarding these studies as a package (i.e., that the link between the needs for input and psychological experiences changes depending on input deprivation duration) must be taken with caution. Under ideal circumstances, Studies 3 and 4 would have been combined into a single study with a 2 (duration: long vs. short term) x 2 (deprivation: absent vs. present) between-subjects design. In other words, this study would consist of 4 conditions: one in which people would be deprived of input long-term, one in which they would be deprived of input short-term, one in which they would be exposed to input long-term, and one in which they would be exposed to input short-term. Based on this study, one could then analyse both whether the link between the needs for social, material, and sensation seeking input and various behaviours or emotional experiences differs as a function of deprivation duration and presence vs. absence of input.

However, a closer look at this “ideal” design reveals that it would face similar problems as the current design. First, given that the conditions in which external input is manipulated long-term involve a longitudinal design, participant attrition would make it difficult to compare these conditions with the ones in which the input is manipulated short-term. Indeed, it would be difficult to conclude whether any differences occurred due to the experimental manipulation or due to unequal participant samples caused by the attrition. Moreover, given that the long-term conditions would require substantially more time than the short-term conditions, it would be unclear whether any differences between them occurred due to the experimental manipulation or due to some unknown factor that took place over time.

When it comes to comparing the conditions in which absence versus presence of external input is manipulated, various methodological problems would also arise. Most importantly, it

would be difficult to understand whether an effect is created by a presence versus absence of social, material, or sensation seeking input. It appears that this issue could be resolved by independently manipulating either of these forms of input. However, the main obstacle in this regard is that the forms of input seem to be interlinked, and it is very difficult to manipulate only one of them while keeping others constant. In a pilot study I conducted that will be part of another paper I have not yet published, I presented participants with different scenarios about situations in which I manipulated each of the three input components, and then asked participants to rate the “quantity” of each input component a situation contains. The result showed that in most cases increasing social input at least to some degree increases sensation seeking input and may also increase material input, and it is highly challenging to disentangle the three input components. Overall, based on these considerations, I conclude that even the “ideal” design I describe would not represent an improvement compared to the current design, and a major methodological advancement would need to be undertaken to eliminate the weaknesses from which the present design of Studies 3-4 suffers.”

For these reasons, and because Study 4 tackles one of the two key research questions my paper explores, I would prefer to keep it in the package, and I hope that this comprehensive discussion of its limitations helps provide a deeper insight into the concerns you raise in your comment.